# GenDataAgent: On-the-fly Dataset Augmentation with Synthetic Data

**Zhiteng Li**[1*]**, Lele Chen**[2]**, Jerone T. A. Andrews**[2†]**, Yunhao Ba**[2]**, Yulun Zhang**[1]**, Alice Xiang**[2]

[1]Shanghai Jiao Tong University, [2]Sony AI

https://github.com/SonyResearch/GenDataAgent

## Abstract

We propose a generative agent that augments training datasets with synthetic data for model fine-tuning. Unlike prior work, which uniformly samples synthetic data, our agent iteratively generates relevant samples on-the-fly, aligning with the target distribution. It prioritizes synthetic data that complements difficult training samples, focusing on those with high variance in gradient updates. Experiments across several image classification tasks demonstrate the effectiveness of our approach.

## 1 Introduction

Generative models can produce photo-realistic images from text prompts (Brock et al., 2018; Razavi et al., 2019; Ho et al., 2020; Saharia et al., 2022; Rombach et al., 2022; Sohl-Dickstein et al., 2015; Ramesh et al., 2022; Nichol et al., 2022). These models are increasingly used to replace (Sarıyıldız et al., 2023; Hammoud et al., 2024; Shipard et al., 2023) or augment real data in training datasets (Yuan et al., 2023; Dunlap et al., 2024; Astolfi et al., 2023). This shift is largely motivated by the significant time and labor costs associated with collecting and annotating real data (Tian et al., 2024; Yuan et al., 2023; Sarıyıldız et al., 2023; Besnier et al., 2020; Dunlap et al., 2024).

Despite their promise, synthetic data augmentation techniques often lack feedback mechanisms during downstream model training, potentially reducing sample utility (Hemmat et al., 2023). Approaches such as prompt engineering (Sarıyıldız et al., 2023; Lei et al., 2023; Azizi et al., 2023), image-conditioned generation (Bordes et al., 2022; Blattmann et al., 2022), diffusion inversion (Zhou et al., 2023; Zhao & Bilen, 2022), and low-density region sampling (Um et al., 2024; Sehwag et al., 2022) aim to improve synthetic data quality and coverage. However, discrepancies between synthetic and target data distributions (Shin et al., 2023; He et al., 2022; Borji, 2022), along with limited diversity in generated samples (Hall et al., 2023; Bianchi et al., 2023; Luccioni et al., 2024; Jahanian et al., 2021), continue to undermine their effectiveness.

In this work, we introduce *GenDataAgent*, a generative agent for training dataset augmentation that addresses key challenges in synthetic data generation. Our method dynamically generates high-quality synthetic samples on-the-fly while ensuring alignment with the target training distribution. By prioritizing diverse and informative synthetic data that complement *marginal* real examples—i.e., difficult examples near the decision boundary—through feedback mechanisms, our approach enhances the generalization performance of downstream models fine-tuned on augmented datasets. Unlike prior research (Hemmat et al., 2023; Shao et al., 2024; Ye-Bin et al., 2023; Liu et al., 2020; Kozerawski et al., 2020), GenDataAgent does not rely on distributional assumptions, such as long-tailed data or datasets with few or no examples per class.

**Our key contributions are:**

- GenDataAgent, an on-the-fly generative agent that augments image classification datasets with Stable Diffusion v1.5 (Rombach et al., 2022), ensuring alignment with the target distribution.
- A feedback-driven sampling strategy that prioritizes marginal real examples, filters outliers via variance of gradients (Agarwal et al., 2022), and enhances diversity through Llama-2 (Touvron et al., 2023) text prompt perturbation.

---

* Work done during an internship at Sony AI   † Corresponding author: jerone.andrews@sony.com

- State-of-the-art generalization performance, improved fairness, and empirical evidence that GenDataAgent effectively complements real-world training data.

## 1.1 RELATED WORK

**Synthetic Data Generation.** The use of generative models for training data augmentation has grown, with diffusion-based methods (Bansal & Grover, 2023; He et al., 2022; Shipard et al., 2023; Trabucco et al., 2023; Besnier et al., 2020; Sarıyıldız et al., 2023; Hammoud et al., 2024; Tian et al., 2024; Yuan et al., 2023; Azizi et al., 2023; Hemmat et al., 2023; Dunlap et al., 2024; Astolfi et al., 2023) increasingly replacing generative adversarial networks (Zhao & Bilen, 2022; Li et al., 2022; Zhang et al., 2021; Kumar et al., 2022; Sharmanska et al., 2020). While text-guided diffusion models produce high-quality synthetic images, their effectiveness in training models remains inconsistent due to persistent distributional gaps (Shin et al., 2023; He et al., 2022; Borji, 2022; Hemmat et al., 2023) and limited sample diversity (Hall et al., 2023; Bianchi et al., 2023; Luccioni et al., 2024).

To mitigate these issues, some methods sample from low-density regions (Um et al., 2024; Sehwag et al., 2022; Samuel et al., 2023), capturing rare attributes not well-represented in high-density areas. However, these approaches rely on assumptions about data distributions, such as long-tailed datasets or low-shot classes, and do not explicitly account for the utility of the generated data. In contrast, we use uncalibrated *marginal scores*—i.e., the target model's predicted probability for a given class—to guide synthetic data generation, ensuring it complements marginal real examples.

**Synthetic Data Augmentation.** Prompt engineering with text-conditioned diffusion models has been explored for classification (Sarıyıldız et al., 2023; Lei et al., 2023; Azizi et al., 2023). Sarıyıldız et al. (2023) use manual, class-agnostic prompts to reduce semantic issues (e.g., polysemy) and improve diversity, while Lei et al. (2023) apply automated image captioning. However, these methods do not explicitly ensure the generated data aligns with the target distribution or contributes useful information. As with prior work (Yuan et al., 2023; Sarıyıldız et al., 2023; Lei et al., 2023; He et al., 2022), they produce static, bloated datasets containing redundant and uninformative samples, as they lack feedback from the downstream model.

Hemmat et al. (2023) introduce feedback-guided synthesis but rely on a single offline feedback cycle. In contrast, we apply on-the-fly filtering during training, prioritizing difficult synthetic samples with higher variance in gradient updates (Agarwal et al., 2022) to avoid noisy or unrepresentative data (He et al., 2022; Hemmat et al., 2023; Shin et al., 2023). Additionally, our approach can integrate with domain adaptation techniques (Tang & Jia, 2023) to further narrow distribution gaps.

Another line of work (Li et al., 2023a; Jiang et al., 2021) augments training datasets with real-world data. For example, Li et al. (2023a) retrieve and rank internet images based on their expected reward. While this increases diversity, internet-sourced data raises privacy and copyright concerns (Samuelson, 2023; Andrews et al., 2024; Longpre et al., 2023; Besnier et al., 2020; Metcalf & Crawford, 2016; Orekondy et al., 2018; Birhane & Prabhu, 2021; Birhane et al., 2021).

## 2 GENDATAAGENT: ON-THE-FLY DATASET AUGMENTATION

We introduce GenDataAgent, a generative agent that dynamically augments vision training datasets with synthetic data during model fine-tuning. GenDataAgent generates target distribution-aligned synthetic samples to enhance supervised image classification, prioritizing diverse and informative data that complement marginal real training samples, particularly those with high gradient update variance. This approach improves generalization and fairness. Pseudocode for GenDataAgent is presented in Algorithm 1.

### 2.1 TEXT-TO-IMAGE GENERATOR

Leveraging text-guided diffusion models, we use Stable Diffusion v1.5 (SD) as a text-to-image generator. Following Yuan et al. (2023), to bridge the distribution gap between SD's training data and the target real data, we adapt SD to better align with the target distribution.

Let $\mathcal{T} = \{(x_i, y_i, p_i, c_i)\}_{i=1}^n$ be the target training dataset, where $x_i$ is a real image, $y_i$ its numerical class label, $p_i$ its semantic class name, and $c_i$ its BLIP-2-generated (Li et al., 2023b) "raw" image

---

**Algorithm 1** `GenDataAgent`

---

1: **Input:** target dataset $\mathcal{T} = \{(x_i, y_i, p_i, c_i)\}_{i=1}^N$, pretrained multi-class classfication model $f$, stable diffusion model `SD`, image feature extractor `CLIP`, image captioning model `BLIP-2`, large language enhance model `Llama-2`
2: Generate image features $\Psi$ for real data $\mathcal{T}$ by `CLIP`
3: Adapt `SD` to target distribution with $\mathcal{T}, \mathcal{P} = \{p_i\}_{i=1}^N, \mathcal{C}$ and $\Psi$                          (§2.1)
4: **for** iter $= 1, 2, \ldots$ **do**
5:     **if** iter $<= 3$ **then**     // Stage-1
6:         Fine-tune $f$ **only** on $\mathcal{T}$, save model checkpoints $\{f_i\}_{i=1}^3$                  (§2.4)
7:     **else**               // Stage-2
8:         Sample feedback $\mathcal{M} \subset \mathcal{T}$ of marginal examples                     (§2.2)
9:         Perturb image captions for marginal examples $\mathcal{C}'_{\mathcal{M}} \leftarrow \text{Llama-2}(\mathcal{P}_{\mathcal{M}}, \mathcal{C}_{\mathcal{M}})$     (§2.3)
10:         Generate diverse synthetic data $\mathcal{S}_c \leftarrow \text{GenData}_{x \in \mathcal{M}}^m(x, \mathcal{C}'_x = \{c'_{x,i}\}_{i=1}^m)$
11:         **for** $j = 1, \ldots, M$ **do**     // traverse all categories
12:             Compute VoG score for each synthetic data $x_i \in \mathcal{S}_c$
13:             Filter out images of $j$-th class $\mathcal{S}_{f,j} \leftarrow \underset{x_i \in \mathcal{S}_{f,j} \subset \mathcal{S}_c}{\arg\min} \sum_i \text{VoG}_i$              (§2.4)
14:         **end for**
15:         Combine real and synthetic data as training data $\mathcal{D} \leftarrow \mathcal{T} \cup (\mathcal{S}_c \backslash \bigcup_j \mathcal{S}_{f,j})$
16:         Fine-tune $f$ on the combined dataset $f_{\theta_{\text{iter}+1}} \leftarrow f_{\theta_{\text{iter}}}(\mathcal{D})$
17:     **end if**
18: **end for**

---

caption. The mean feature vector of class $y_i$, extracted using CLIP (Radford et al., 2021), is denoted as $\mu_{y_i}$. Yuan et al. (2023) propose constructing text prompts by concatenating semantic class names $p_i$ and image captions $c_i$, along with $\psi_{y_i}$, which provides visual guidance by estimating intra-class feature distributions. We adopt this approach to fine-tune SD with LoRA (Hu et al., 2021), using the multi-modal prompt: `a photo of` $p_i$, `which is` $c_i$, $\psi_{y_i}$.

For text-to-image generation, we prioritize sample quality over diversity by setting the prompt guidance value to 7.5, whereas prior methods (Sarıyıldız et al., 2023; Yuan et al., 2023) typically use 2. This choice is driven by our improved approach to introducing sample diversity without compromising quality (Appendix A).

## 2.2 GENERATOR FEEDBACK VIA MARGINAL SAMPLES

Previous work (Yuan et al., 2023; Sarıyıldız et al., 2023) generates synthetic data aligned with the target distribution but overlooks its utility, raising concerns about its effectiveness. Inspired by sample reweighting techniques (Freund & Schapire, 1995; Johnson & Khoshgoftaar, 2019; Ren et al., 2018), we emphasize the importance of synthetic data that complement real samples near the model's decision boundary. Exposing the model to these critical regions of the data space can enhance generalization, even if it initially increases losses.

To achieve this, we use a *marginal score*, defined as the target model's predicted probability for class $y_i$ given input $x_i$. Specifically, the marginal score is computed as

$$\hat{p}(y = y_i \mid x_i) = \frac{\exp(z_{y_i}^i)}{\sum_j \exp(z_j^i)}, \tag{1}$$

where $z_j$ denotes the logit for class $j$. We rank all samples in the target dataset by their marginal scores and select the $k$ lowest-scoring samples as feedback for guiding synthetic data generation. This set of $k$ marginal samples is denoted as $\mathcal{M}$.

Figure 1 illustrates clustering patterns in the pretrained target model's feature space, showing that instances with high marginal scores form clearer boundaries, while those with low scores exhibit greater diversity and noise. High-scoring images tend to be centrally positioned, with representative object depictions, whereas low-scoring images are more challenging to classify. Moreover, empirical results in Section 3.3 confirm that using predicted probabilities is more effective than entropy-based selection (Hemmat et al., 2023).

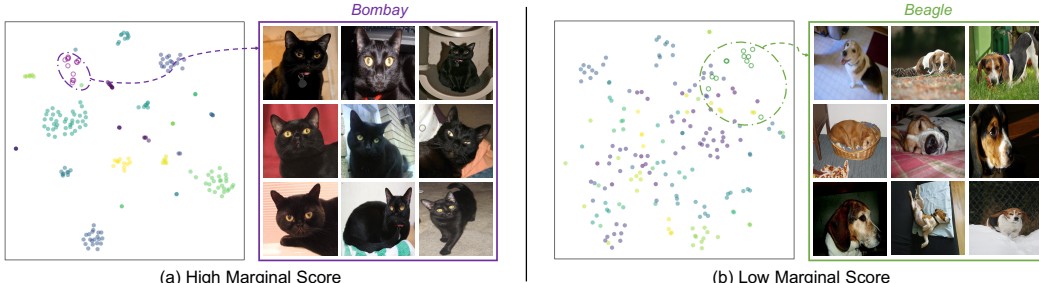

Figure 1: Feature space visualization of Oxford-IIIT Pets samples with the highest (Top-200) and lowest (Bottom-200) marginal scores. Insets show examples from high-scoring (`Bombay`) and low-scoring (`Beagle`) clusters, with colors representing distinct classes.

## 2.3 ENHANCING DIVERSITY VIA CAPTION PERTURBATIONS

Images generated with identical captions but different seeds often exhibit limited diversity, even with a high guidance value (Figure 2). This redundancy reduces the effectiveness of synthetic data augmentation (Hall et al., 2023; Bianchi et al., 2023). To mitigate this, we use Llama-2 (Touvron et al., 2023), a large language model, to modify image captions. However, since Llama-2 tends to produce lengthy descriptions, which may introduce excessive outliers, we constrain its outputs to short sentences using a word limit. We achieve this by providing Llama-2 with the following tailored prompts:

```
role : system, content: You are an editor tasked with subtly altering
the scene described after a comma in a sentence. The goal is to change
the scene slightly in no more than 10 words. Respond with m versions.

role : user, content: Given the sentence "a photo of pi, which is ci",
slightly alter the scene described after the comma to depict a similar
yet different scenario.
```

Examples are provided in Figure 2 and Appendix A. To maintain consistency, we use the same prompt format—combining class name $p_i$ and BLIP-2-generated raw image caption $c_i$—for both adapting SD (Section 2.1) and generating synthetic data. This ensures closer alignment between synthetic and real data distributions.

Given the set $\mathcal{M}$ of marginal examples from the classifier, we generate $m$ perturbed captions for each $x \in \mathcal{M}$ using Llama-2:

$$\mathcal{C}'_x = \text{CaptionPerturbation}^m(p_x, c_x), \tag{2}$$

where $\mathcal{C}' = \{\mathcal{C}'_x \mid x \in \mathcal{M}\}$ represents the full set of perturbed captions. Using these perturbations, we generate synthetic data as follows:

$$\mathcal{S}_c = \text{GenData}^m_{x \in \mathcal{M}}(x, \mathcal{C}'_x = \{c'_{x,i}\}^m_{i=1}), \tag{3}$$

where $m$ denotes the number of caption variations per image. The resulting synthetic dataset size $|\mathcal{S}_c| = k \cdot m$ can be scaled to match different augmentation ratios, such as $1 \cdot |\mathcal{T}|$ or $10 \cdot |\mathcal{T}|$.

As shown in Figure 2, Llama-2 perturbations introduce subtle yet meaningful variations while preserving core semantics. For example, in the Oxford-IIIT Pets dataset (Parkhi et al., 2012), an `Abyssinian` cat's posture or viewpoint may shift slightly, but the overall scene remains consistent. This ensures perturbations target marginal examples rather than generic augmentations. In the feature space, synthetic images from raw captions cluster closely with real data, whereas those from Llama-2 perturbations exhibit greater spread, enhancing diversity while maintaining class relevance.

## 2.4 IN-DISTRIBUTION DATA GENERATION

Synthetic data generated by the agent may include out-of-distribution samples (Figure 3c), which can introduce noise and degrade model performance. To mitigate this, we employ variance of

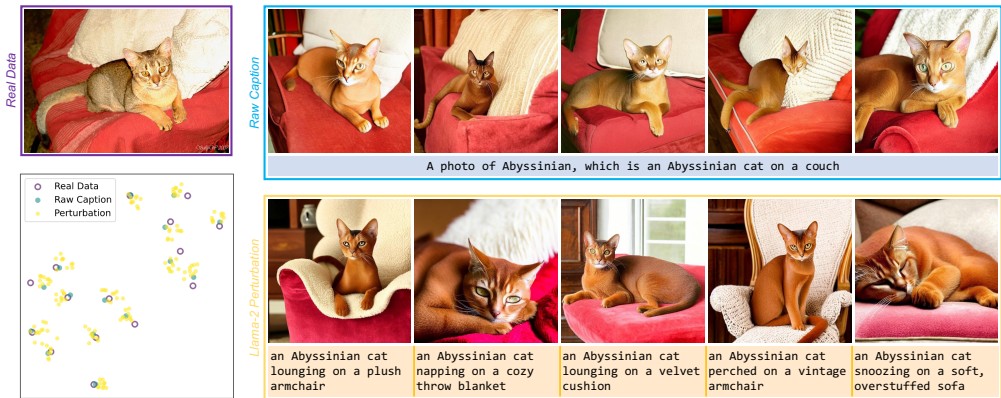

Figure 2: Comparison of real images, synthetic images generated with identical captions, and synthetic images generated with Llama-2 perturbed captions. The lower-left t-SNE visualization shows class clusters in feature space. For brevity, the perturbed captions omit the prefix: `a photo of` $p_i$. All synthetic data is generated using a prompt guidance value of 7.5.

gradients (VoG) (Agarwal et al., 2022), a filtering mechanism inspired by prior work on sample hardness. *Harder negatives*—samples inducing larger gradient updates—tend to be more informative for learning (Li et al., 2023a; Robinson et al., 2020; Ge, 2018; Schroff et al., 2015).

Classifier fine-tuning follows two stages: (1) in the initial fine-tuning stage ($N = 3$ iterations, 10 epochs each), the agent trains the model on the target dataset $\mathcal{T}$, saving checkpoints at each iteration; (2) in the refinement stage ($N > 3$), the agent uses model feedback and checkpoints to improve data selection.

After generating synthetic data $\mathcal{S}_c$, the agent computes the logit gradient w.r.t. each pixel in $x_i$:

$$G_i = \frac{\partial z^i_{y_i}}{\partial x_{i,d}}, \tag{4}$$

where $d = \{1, 2, \ldots, W\}$ and $W$ is the total number of pixels in $x_i$. Each tuple $(x_i, y_i) \in \mathcal{S}_c$ is evaluated using VoG, which measures gradient variance across training checkpoints:

$$\mu_i = \frac{1}{3}(G_i^{10} + G_i^{20} + G_i^{30}), \tag{5}$$

$$\text{VoG}_i = \sqrt{\frac{1}{3}[(G_i^{10} - \mu_i)^2 + (G_i^{20} - \mu_i)^2 + (G_i^{30} - \mu_i)^2]}, \tag{6}$$

where $G_i^{\eta}$ is the gradient at epoch $\eta$. Experiments (Table 5) show minimal performance differences when using 3, 4, or 5 checkpoints, so we adopt 3 checkpoints for efficiency.

Since in-distribution samples exhibit higher VoG values—initially receiving large gradients that decrease as the model learns—GenDataAgent filters out out-of-distribution samples by removing the lowest-VoG data per class $j$:

$$\mathcal{S}_{f,j} = \arg\min_{x_i \in \mathcal{S}_{f,j} \subset \mathcal{S}_c} \sum_i \text{VoG}_i \quad \text{s.t.} \quad |\mathcal{S}_{f,j}| = o_j. \tag{7}$$

Even after adapting SD to the target dataset, outliers persist in synthetic data (Figure 3, Appendix B). However, VoG filtering effectively removes them, as t-SNE visualizations (Figure 3) show: filtered synthetic data centers around real data, while VoG outliers remain entangled with noise.

While caption perturbation enhances synthetic data diversity, it can introduce more outliers. However, VoG filtering effectively removes most outliers, balancing diversity and in-distribution alignment.

### 2.5 ON-THE-FLY FEEDBACK AND FINE-TUNING

Prior work (Yuan et al., 2023; Sarıyıldız et al., 2023; Lei et al., 2023; He et al., 2022) augments real datasets with synthetic data statically, ignoring classifier feedback. LDM-FG (Hemmat et al., 2023) incorporates classifier-driven feedback but only applies it in a *single* offline cycle.

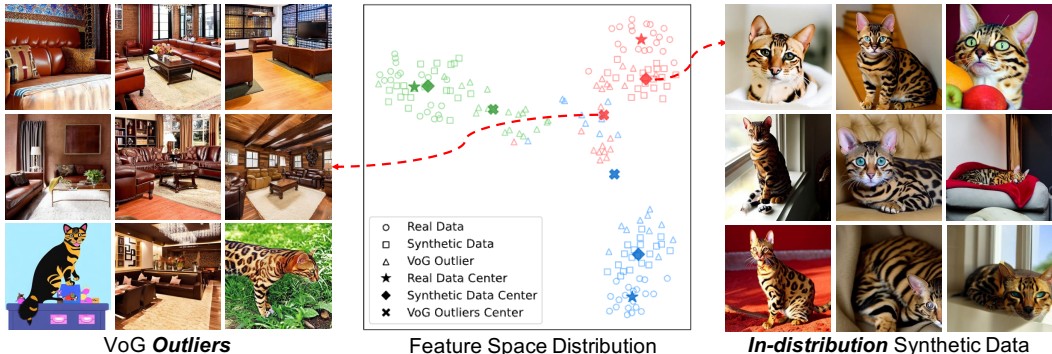

VoG **Outliers**          Feature Space Distribution          **In-distribution** Synthetic Data

Figure 3: t-SNE visualization of VoG filtering on the Oxford-IIIT Pets dataset. For each class, 20 samples are randomly selected from real data, filtered synthetic data, and synthetic samples with the bottom 25% VoG scores.

In contrast, our approach integrates on-the-fly feedback during training, ensuring dynamic adaptation. As outlined in Algorithm 1, in each stage 2 iteration, we: (i) resample marginal examples based on the model's current state, (ii) send this feedback to GenDataAgent, which generates new synthetic data, and (iii) fine-tune the classifier on a combination of real and synthetic data. That is:

$$f_{\theta_{\text{iter}+1}} \leftarrow f_{\theta_{\text{iter}}}(\mathcal{T} \cup (\mathcal{S}_c \setminus \bigcup_j \mathcal{S}_{f,j})), \tag{8}$$

where $f_{\theta_{\text{iter}}}$ is the model from the previous iteration. This iterative loop refines both data selection and model training. Each new iteration updates $f_{\theta_{\text{iter}+1}}$, resampling marginal examples and generating new feedback for GenDataAgent. This ensures synthetic data remains informative, task-relevant, and distribution-aligned throughout training.

## 3 EXPERIMENTS

We evaluate GenDataAgent in a supervised learning setting, following prior work (Yuan et al., 2023; Sarıyıldız et al., 2023; He et al., 2022). Our experiments cover two scenarios: (i) training a classifier using synthetic data alone, and (ii) using synthetic data to augment real data. In both cases, we use ResNet-50, while the second scenario also includes evaluations with CLIP ResNet-50 and CLIP ViT. Additional details and experiments are provided in Appendices C to F.

### 3.1 EXPERIMENTAL SETUP

**Synthetic Data Only.**   We assess whether synthetic data can replace real training data in fine-tuning an image classifier. Since GenDataAgent requires real data $\mathcal{T}$ for feedback, we replace $\mathcal{T}$ with an equivalent number of synthetic samples and iteratively inject augmented synthetic data. We compare our method against Real-Fake (Yuan et al., 2023), ImageNet-Clone (Sarıyıldız et al., 2023), CiP (Lei et al., 2023), and SyntheticData (He et al., 2022).

**Synthetic Data Augmentation.**   We compare GenDataAgent with the state-of-the-art Real-Fake and Internet Explorer (Li et al., 2023a). To ensure fairness in comparison with Internet Explorer, we maintain our distribution adaptation and Llama-2 perturbations while replacing marginal sampling and VoG filtering with the 15-NN similarity approach.

**Datasets.**   We evaluate GenDataAgent on ImageNet-100 (IN100) (Tian et al., 2020) and five fine-grained datasets: Oxford-IIIT Pets (Parkhi et al., 2012), Flowers-102 (Nilsback & Zisserman, 2008), Birdsnap (Berg et al., 2014), CUB-200-2011 (Wah et al., 2011), and Food-101 (Bossard et al., 2014).

Following prior work, we use ImageNet-pretrained backbone models for all datasets except IN100, where classifiers are trained from scratch, as in Real-Fake.

Table 1: Top-1 accuracy / worst-case disparity for image classification using only synthetic data. **Bold** values indicate the best performance.

| Model | Pets | CUB | Flowers | Birdsnap | Food | IN100 |
|---|---|---|---|---|---|---|
| *Training with **only synthetic data*** | | | | | | |
| ImageNet-Clone[‡] | 79.7/0.20 | 29.4/0.00 | 27.2/0.00 | 24.1/0.00 | 55.6/0.10 | 64.2/0.20 |
| CiP[‡] | 86.5/0.16 | 35.1/0.00 | 25.8/0.00 | 30.7/0.00 | 55.5/0.15 | 64.7/0.20 |
| SyntheticData[†] | 86.8/− | 56.9/− | 67.1/− | 38.1/− | 80.4/− | −/− |
| Real-Fake[‡] | 89.5/0.40 | 66.0/0.00 | 66.9/0.00 | 54.2/0.00 | 80.1/**0.42** | 82.8/**0.40** |
| GenDataAgent (Ours) | **90.3**/**0.44** | **71.4**/0.00 | **76.9**/**0.11** | **54.7**/0.00 | **81.2**/**0.42** | **87.2**/**0.40** |

[†] SyntheticData uses the CLIP framework but retains the ResNet-50 backbone.   [‡] ImageNet-Clone, CiP, and Real-Fake results are reproduced for all datasets.

Table 2: Top-1 accuracy / worst-case disparity for image classification with synthetic data augmentation. **Bold** values indicate the best performance. $\Delta$ represents the improvement of GenDataAgent over the only real data setup, with gains highlighted in green.

| Model | Pets | CUB | Flowers | Birdsnap | Food | IN100 |
|---|---|---|---|---|---|---|
| *ResNet-50 backbone* | | | | | | |
| Only real data | 93.6/0.40 | 83.1/0.13 | 87.4/0.40 | 73.0/0.00 | 86.8/0.63 | 87.4/0.20 |
| Real-Fake[‡] | 94.2/0.48 | 83.1/0.00 | 89.0/0.50 | 73.0/0.00 | 87.4/0.61 | 88.6/**0.40** |
| Internet Explorer | 94.5/0.52 | 83.6/**0.25** | 90.2/**0.56** | 73.9/0.00 | 87.3/0.63 | 88.9/**0.40** |
| GenDataAgent (Ours) | **94.7**/**0.56** | **83.9**/**0.25** | **91.0**/**0.56** | **74.5**/0.00 | **87.8**/**0.64** | **90.1**/**0.40** |
| $\Delta$ with only real data | +1.1/0.16 | +0.8/0.12 | +3.6/0.16 | +1.5/0.00 | +1.0/0.01 | +2.7/0.20 |
| *CLIP ResNet-50 backbone* | | | | | | |
| Only real data | 77.8/0.28 | 66.3/0.00 | 69.0/0.24 | 64.6/0.00 | 82.2/0.47 | 87.0/0.20 |
| Real-Fake[‡] | 80.7/**0.32** | 66.9/0.00 | 71.0/0.24 | 65.7/0.00 | 86.1/0.61 | 88.0/**0.40** |
| Internet Explorer | 81.3/**0.32** | 67.7/0.00 | 72.2/0.24 | 66.2/0.00 | 86.3/0.61 | 88.4/**0.40** |
| GenDataAgent (Ours) | **82.0**/**0.32** | **68.2**/0.00 | **72.8**/**0.30** | **66.7**/0.00 | **86.5**/**0.63** | **89.1**/**0.40** |
| $\Delta$ with only real data | +4.2/0.04 | +1.9/0.00 | +3.8/0.06 | +2.1/0.00 | +4.3/0.16 | +2.1/0.20 |
| *CLIP ViT backbone* | | | | | | |
| Only real data | 92.1/0.40 | 80.5/0.00 | 86.5/0.33 | 65.8/0.00 | 54.4/0.15 | 63.2/0.20 |
| Real-Fake[‡] | 92.8/0.40 | 80.7/0.00 | 94.9/0.60 | 67.8/0.00 | 63.7/0.20 | 64.9/0.20 |
| Internet Explorer | 92.9/0.40 | 81.8/0.00 | 95.5/0.67 | 68.4/0.00 | 65.1/0.24 | 65.9/0.20 |
| GenDataAgent (Ours) | **93.3**/**0.48** | **82.6**/**0.13** | **96.1**/**0.78** | **69.6**/0.00 | **67.0**/**0.26** | **66.3**/0.20 |
| $\Delta$ with only real data | +1.2/0.08 | +2.1/0.13 | +9.6/0.45 | +3.8/0.00 | +12.6/0.11 | +3.1/0.00 |

[‡] Real-Fake results are reproduced for all datasets.

**Evaluation Metrics.** Following (Yuan et al., 2023; Sarıyıldız et al., 2023), we evaluate GenDataAgent using top-1 accuracy and worst-case disparity (min-max accuracy ratio) (Ghosh et al., 2021) to assess fairness. As our sampling strategy targets marginal examples, we expect improved classifier fairness, a key factor in real-world applications.

## 3.2 QUANTITATIVE RESULTS

**Synthetic Data Only.** Table 1 presents the top-1 classification accuracy and worst-case disparity for various methods across different datasets in the ***synthetic-only*** setup. GenDataAgent significantly outperforms all other methods in classification accuracy, demonstrating that our on-the-fly generation approach is effective even without a real dataset to initialize the feedback mechanism. Notably, GenDataAgent achieves performance levels comparable to models trained exclusively on real data, particularly on the IN100 dataset. (Refer to Appendix C for the mean and standard deviation of multiple runs.)

**Synthetic Data Augmentation.** Table 2 presents results for the ***real + synthetic*** setup with different pretrained backbones. GenDataAgent consistently improves classification accuracy and worst-case disparity over the ***only real data*** setup across all benchmarks and outperforms all competing methods across all backbone architectures.

Comparing GenDataAgent with Internet Explorer highlights the effectiveness of marginal sampling and VoG filtering. As shown in Table 2, synthetic data generated on the fly (e.g., by our method and Internet Explorer) improves model fairness, reducing worst-case disparity. In contrast, Real-Fake increases worst-case disparity in the CUB and Food datasets when using the ResNet-50 backbone. This suggests that simply generating synthetic data can amplify existing class biases if it does not

Table 3: Ablation study of marginal sampling, Llama-2 caption perturbations, and VoG filtering. Top-1 accuracy / worst-case disparity results are reported. **Bold** values indicate the best performance.

| Marginal Sampling | Perturbation | VoG Filtering | Pets | CUB | Flowers | Birdsnap | Food | IN100 |
|---|---|---|---|---|---|---|---|---|
| - | - | - | 93.6 / 0.40 | 83.1 / 0.13 | 87.4 / 0.40 | 73.0 / **0.00** | 86.8 / 0.63 | 87.4 / 0.20 |
| | | | 94.2 / 0.48 | 83.1 / 0.00 | 89.0 / 0.50 | 73.0 / **0.00** | 87.4 / 0.61 | 88.6 / **0.40** |
| ✓ | | | 94.4 / **0.56** | 83.6 / **0.25** | 90.0 / 0.50 | 73.6 / **0.00** | 87.4 / 0.63 | 89.1 / **0.40** |
| ✓ | ✓ | | 94.6 / **0.56** | 83.9 / **0.25** | 90.1 / 0.50 | 73.6 / **0.00** | 87.8 / 0.64 | 89.9 / **0.40** |
| ✓ | ✓ | ✓ | **94.7** / **0.56** | **83.9** / **0.25** | **91.0** / **0.56** | **74.5** / **0.00** | **87.8** / **0.64** | **90.1** / **0.40** |

Table 4: Ablation study on feedback criteria (Section 2.2). **Bold** values indicate the best performance.

| Feedback | Pets | CUB | Flowers | Birdsnap | Food | IN100 |
|---|---|---|---|---|---|---|
| Entropy Score | 94.6 / 0.52 | 83.2 / 0.13 | **91.0** / **0.56** | 73.6 / 0.00 | 86.9 / 0.58 | 89.4 / 0.40 |
| Marginal Score | **94.7** / **0.56** | **83.9** / **0.25** | **91.0** / **0.56** | **74.5** / **0.00** | **87.8** / **0.64** | **90.1** / **0.40** |

account for biases present in the original real dataset. Our on-the-fly approach mitigates this issue by dynamically interacting with the model during training. However, in the ***synthetic-only*** setting, where there is no initial real dataset for feedback, both GenDataAgent and Real-Fake experience performance drops in worst-case disparity.

## 3.3 ABLATION STUDY

**Component-Wise Ablation.** We assess the impact of marginal sampling (Section 2.2), Llama-2 caption perturbations (Section 2.3), and VoG filtering (Section 2.4) in Table 3. Starting with static synthetic data augmentation, we incrementally add each component, observing consistent improvements across datasets. (For additional hyperparameter ablation analysis, see Appendix D.)

**Feedback Criteria.** We compare marginal score-based sampling with an entropy-based criterion (Hemmat et al., 2023; Kolossov et al., 2024) in Table 4. Results show that entropy performs similarly or worse, while marginal score-based sampling is simpler and more efficient. Thus, we adopt marginal sampling as the feedback criterion.

**Number of VoG Checkpoints.** We evaluate VoG filtering with 3, 4, and 5 checkpoints (Table 5). Results show minimal performance differences, so we adopt 3 checkpoints to optimize resource efficiency.

Table 5: Effect of VoG checkpoint count.

| #ckpts | Pets | CUB | Flowers | Birdsnap |
|---|---|---|---|---|
| 3 | 94.7/0.56 | 83.9/0.25 | 91.0/0.56 | 74.5/0.00 |
| 4 | 94.5/0.56 | 84.1/0.25 | 91.5/0.56 | 74.4/0.00 |
| 5 | 94.4/0.52 | 84.0/0.25 | 91.4/0.56 | 73.6/0.00 |

**Comparison with Traditional Data Augmentation.** We compare GenDataAgent with RandAugment (Cubuk et al., 2020), a transformation-based augmentation method (Table 6). Results show that traditional augmentation provides minimal gains, whereas GenDataAgent consistently improves both top-1 accuracy and worst-case disparity, highlighting the advantages of synthetic data augmentation.

**Scaling Ablation and Time Analysis.** We examine real-to-synthetic data ratios, comparing Real-Fake (1:10) with GenDataAgent (1:0.5, 1:0.1). The search space of Real-Fake (1:10) matches GenDataAgent (1:0.5), as both generate the same total synthetic data. We also analyze time consumption to assess efficiency and identify bottlenecks.

As shown in Figure 4, blindly increasing synthetic data can degrade performance—Real-Fake (1:10) underperforms Real-Fake (1:1), especially on fine-grained datasets. In contrast, GenDataAgent (1:0.5) surpasses Real-Fake (1:1), demonstrating its scalability, while GenDataAgent (1:0.1) performs on par or better, showing adaptability with fewer synthetic samples. Training with only real data is fastest, while both Real-Fake and GenDataAgent require time to adapt SD to the target distribution. Though GenDataAgent (1:1) is slowed by synthetic data generation, GenDataAgent (1:0.1) balances performance and efficiency, with further improvements possible through more efficient generation models. (GenDataAgent also supports an offline mode for speed-critical scenarios—see Appendix G.)

Table 6: Comparison of RandAugment and GenDataAgent for data augmentation. **Bold** values indicate the best performance.

| Method | Pets | CUB | Flowers | Birdsnap | Food | IN100 |
|---|---|---|---|---|---|---|
| Only real data | 93.6/0.40 | 83.1/0.13 | 87.4/0.40 | 73.0/0.00 | 86.8/0.63 | 87.4/0.20 |
| RandAugment | 93.7/0.40 | 83.0/0.13 | 87.5/0.40 | 73.5/0.00 | 87.0/0.63 | 86.8/0.40 |
| GenDataAgent (Ours) | **94.7**/**0.56** | **83.9**/**0.25** | **91.0**/**0.56** | **74.5**/0.00 | **87.8**/**0.64** | **90.1**/0.40 |

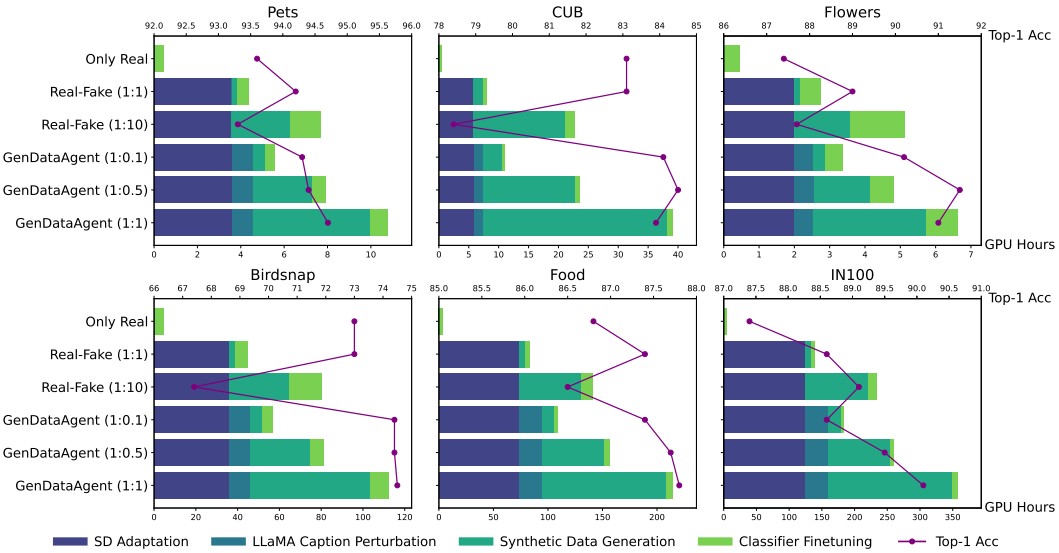

Figure 4: GPU hours per step (bars, left) and top-1 accuracy (line, right) for different real-to-synthetic data ratios. Increasing the synthetic data ratio (e.g., Real-Fake 1:1 to 1:10) can degrade performance, whereas GenDataAgent improves across both small and large ratios.

## 4 ANALYSIS OF GENERATED CONTENT

**Does synthetic data volume impact accuracy?** Figure 5 shows that GenDataAgent-generated synthetic data aligns with training accuracy trends. After training on real data only (stage 1 in Algorithm 1), GenDataAgent generates more synthetic samples for lower-accuracy categories, suggesting awareness of classification accuracy. The incremental view (right side of Figure 5) further shows that $\Delta$ top-1 accuracy from the first iteration to convergence closely follows the increase in synthetic data, indicating a strong correlation between performance gains and synthetic data volume.

**Does synthetic augmentation reduce overfitting?** Figure 6 (a) compares training and validation accuracy after convergence. Unlike static augmentation, our on-the-fly approach reduces the train-validation gap (yellow to red region), indicating improved generalization.

**Do real and synthetic data impact training equally?** No. As shown in Figure 6 (b), we compute the average gradient magnitude of real and synthetic data per iteration to measure their impact. While both gradients decrease as training progresses, the real data gradient decays faster. A distinct boundary separates the process into two stages: early training is dominated by real data, enhancing the model's expressive capacity, while synthetic data plays a larger role in later fine-tuning.

**Does the model treat synthetic data differently from real data during fine-tuning?** Yes. Figure 6 (c) shows that in Real-Fake's static augmentation, synthetic and real data have similar training accuracy but differ significantly from the validation set. In contrast, in on-the-fly augmentation, synthetic data accuracy is notably lower than real data but closely matches the validation set. This suggests that in the static setting, the model overfits synthetic data, whereas in on-the-fly augmentation, synthetic data serves a distinct role in improving generalization.

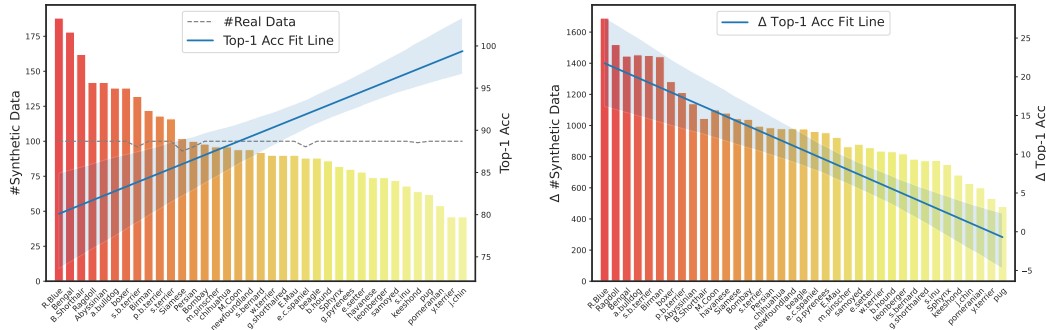

Figure 5: Relationship between synthetic data volume (bar chart) and training top-1 accuracy on the Oxford-IIIT Pets dataset. **Left**: Absolute relation in the first on-the-fly iteration. **Right**: Incremental relation from the first iteration to convergence.

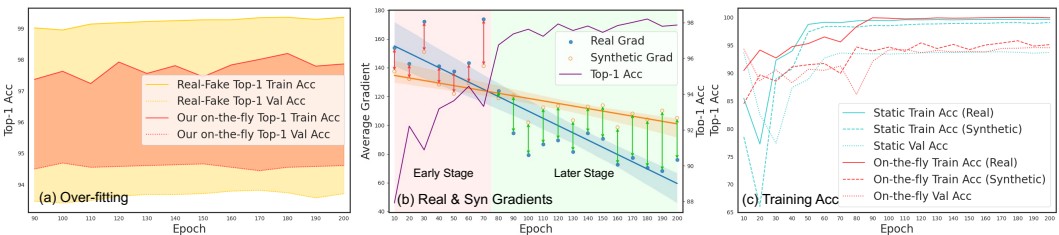

Figure 6: Analysis on the Pets dataset. **(a)** Training and validation accuracy after convergence for Real-Fake and our method. The train-validation gaps are highlighted in yellow (Real-Fake) and red (GenDataAgent). **(b)** Average gradient magnitude of real and synthetic data, with training top-1 accuracy over iterations. Red arrows indicate when real data has a larger impact; green arrows indicate when synthetic data dominates. **(c)** Training accuracy of real and synthetic data in static and on-the-fly settings.

## 5 CONCLUSION

We introduce GenDataAgent, an on-the-fly framework for synthetic data augmentation in computer vision. GenDataAgent first fine-tunes Stable Diffusion to align synthetic data with the target distribution, then prioritizes samples that complement marginal real examples, narrowing the search space. Additionally, Llama-2 caption perturbations enhance diversity, while VoG filtering maintains distribution alignment. Extensive evaluations across image classification tasks demonstrate GenDataAgent's effectiveness, achieving state-of-the-art generalization and improved fairness. Furthermore, our content analysis highlights its potential to drive future advancements in synthetic data techniques.

### ACKNOWLEDGMENTS

This work was funded by Sony Research. ZL and YZ acknowledges further support from Shanghai Municipal Science and Technology Major Project (2021SHZDZX0102) and the Fundamental Research Funds for the Central Universities.

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

## A  EXTENDED COMPARISON OF LLAMA-2 CAPTION PERTURBATIONS

`A photo of` $p_i$ is included in the prompt because BLIP-2, the captioning model, may fail to recognize fine-grained categories. As illustrated in Figure 7 identical captions with different seeds produce similar images at guidance 7.5. While lowering guidance to 2.0 increases diversity, it significantly reduces image quality. In contrast, Llama-2 caption perturbations with guidance 7.5 generate diverse yet high-quality synthetic images.

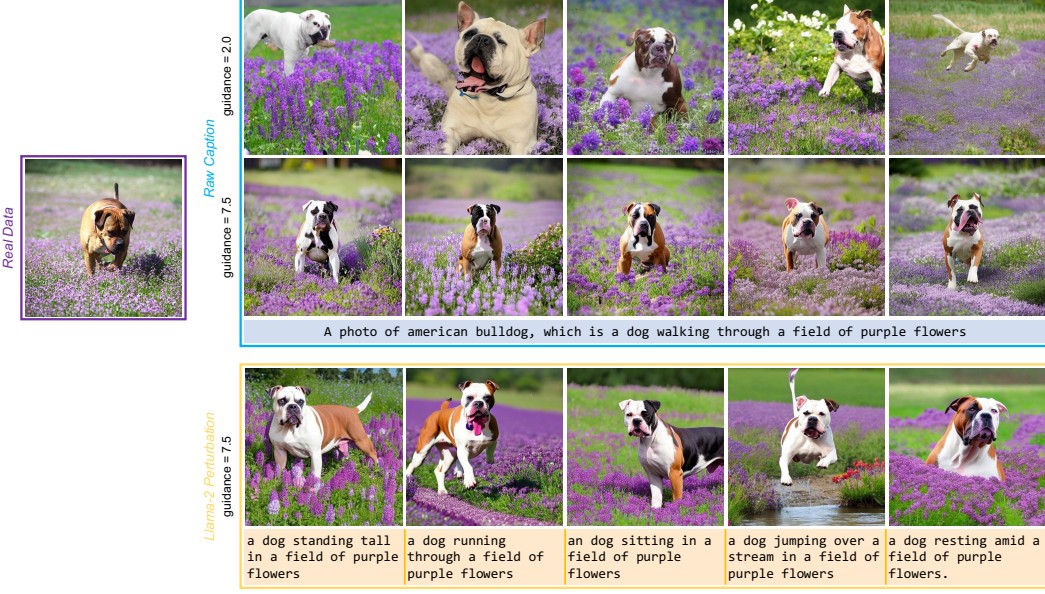

Figure 7: Comparison of raw captions at guidance 2.0 and 7.5, and Llama-2 caption perturbations at guidance 7.5. For brevity, the perturbed captions omit the prefix: `a photo of` $p_i$.

## B  ADDITIONAL VOG FILTERING EXAMPLES

Figure 8 presents additional comparisons of in-distribution synthetic data and VoG-detected outliers across all datasets.

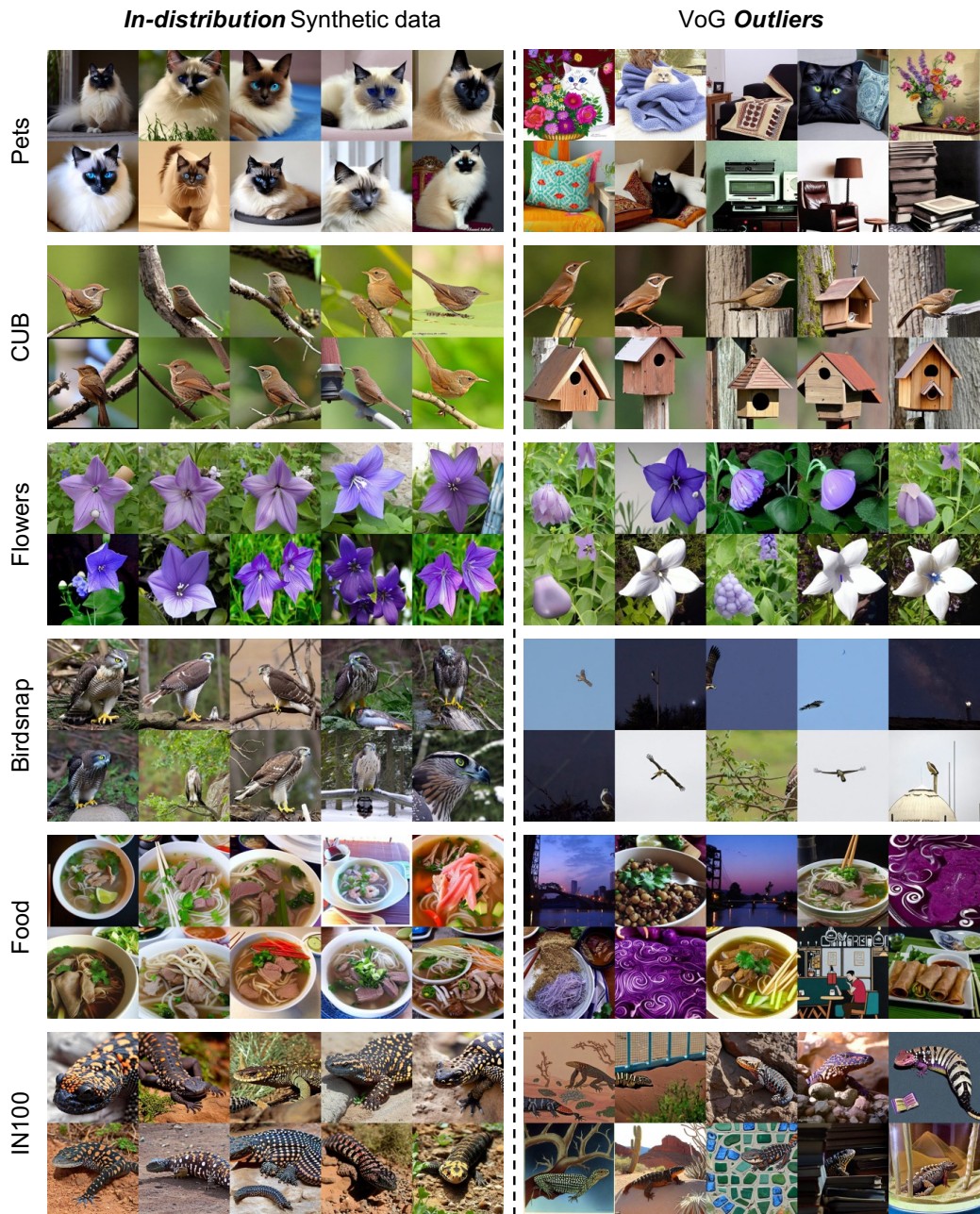

Figure 8: Comparison of in-distribution synthetic data and VoG-detected outliers across all datasets.

## C    REPRODUCIBILITY ACROSS RANDOM SEEDS

We evaluate synthetic data augmentation across three random seeds, as summarized in Table 7.

## D    IMPACT OF THRESHOLDS ON PERFORMANCE

Due to space constraints, additional threshold analysis experiments are presented here.

**Image Strength.**    Following Real-Fake (Yuan et al., 2023), we combine the latent prior of real images to generate synthetic data, with image strength controlling the noise added to the reference image. Higher image strength increases noise in the latent codes. We evaluate thresholds of 0.50, 0.75,

Table 7: Mean and standard deviation of top-1 accuracy and worst-case disparity across three random seeds using a ResNet-50 backbone. **Bold** values indicate the best performance.

| Model | Pets | CUB | Flowers | Birdsnap |
|---|---|---|---|---|
| *Top-1 Accuracy* | | | | |
| Real-Fake[†] | $94.0 \pm 0.15$ | $80.5 \pm 2.25$ | $89.2 \pm 0.32$ | $73.1 \pm 0.31$ |
| Internet Explorer (15-NN similarity)[‡] | $94.3 \pm 0.25$ | $83.2 \pm 0.36$ | $90.7 \pm 0.44$ | $73.7 \pm 0.15$ |
| GenDataAgent (Ours) | $\mathbf{94.6 \pm 0.10}$ | $\mathbf{84.1 \pm 0.17}$ | $\mathbf{91.5 \pm 0.46}$ | $\mathbf{74.0 \pm 0.57}$ |
| *Worst-Case Disparity* | | | | |
| Real-Fake[†] | $0.48 \pm 0.00$ | $0.00 \pm 0.00$ | $0.50 \pm 0.00$ | $0.00 \pm 0.00$ |
| Internet Explorer (15-NN similarity)[‡] | $0.51 \pm 0.02$ | $0.17 \pm 0.07$ | $0.54 \pm 0.03$ | $0.00 \pm 0.00$ |
| GenDataAgent (Ours) | $\mathbf{0.55 \pm 0.02}$ | $\mathbf{0.25 \pm 0.00}$ | $\mathbf{0.56 \pm 0.00}$ | $0.00 \pm 0.00$ |

[†] Results reproduced from (Yuan et al., 2023).    [‡] 15-NN similarity approach from (Li et al., 2023a).

Table 8: Top-1 accuracy / worst-case disparity for different image strength thresholds.

| Image Strength | Pets | CUB | Flowers | Birdsnap | Food | IN100 |
|---|---|---|---|---|---|---|
| 0.50 | 94.0 / 0.44 | 81.0 / 0.25 | **89.9 / 0.50** | 71.4 / 0.00 | 87.4 / 0.63 | 88.6 / 0.40 |
| 0.75 | 94.0 / 0.44 | 82.2 / 0.25 | 88.4 / 0.40 | **73.6 / 0.00** | **87.4 / 0.63** | 88.7 / 0.40 |
| 0.90 | **94.4 / 0.56** | **83.6 / 0.25** | 88.8 / 0.40 | 73.5 / 0.00 | 87.4 / 0.61 | **89.1 / 0.40** |

and 0.90 (Table 8), using a default marginal sampling threshold and no VoG filtering. Results show that lower image strength benefits Flowers for consistency with real data, while higher values improve performance on Pets, CUB, and Birdsnap, highlighting the importance of diversity in synthetic data.

**Marginal Sampling.** Table 9 examines the effect of different marginal sampling thresholds without VoG filtering. A subset of real data is sampled as marginal examples, guiding synthetic data generation. While the optimal threshold varies across datasets, all settings outperform Real-Fake and exhibit stable performance.

**VoG Filtering.** Table 10 shows that increasing VoG filtering consistently improves performance over Real-Fake, confirming the necessity of outlier removal in synthetic data augmentation. The preference for higher filtering ratios suggests that synthetic data quality still lags behind real data, pointing to a direction for future improvements.

# E DATASETS AND TRAINING DETAILS

**Dataset Details.** Dataset statistics are summarized in Table 11.

**Training Details.** We use Stable Diffusion v1.5 (Rombach et al., 2022), following Real-Fake. During adaptation, we condition generation on the prompt: `a photo of` $p_i$`, which is` $c_i$`,` $\psi_{y_i}$. Low-Rank Adaptation (LoRA) fine-tuning is applied with the same hyperparameters as (Yuan et al., 2023) with the following negative prompts: `distorted, unrealistic, blurry, out of frame`. For synthetic data augmentation, we use guidance scale 7.5, whereas for synthetic-only training, we use guidance scale 2.0, as suggested by (Sarıyıldız et al., 2023).

Training hyperparameters for the downstream classifier are listed in Table 12.

# F ADDITIONAL EXPERIMENTAL RESULTS

**Robustness to Common Corruptions.** We assess model robustness by introducing common corruptions such as Gaussian Blur and Speckle Noise, following Hendrycks & Dietterich (2019). All models are trained on clean images and evaluated on corrupted images. As shown in Table 13, GenDataAgent consistently outperforms Real-Fake across all datasets and corruption types.

**Train-Validation Accuracy Gap.** To assess overfitting, we report training and validation accuracy along with their gap after convergence in Table 14. Notably, GenDataAgent reduces this gap compared to Real-Fake, suggesting that the synthetic data—particularly with Llama-2 caption perturbations— helps mitigate overfitting by introducing greater diversity.

Table 9: Top-1 accuracy / worst-case disparity for different marginal score sampling thresholds.

| Marginal Score Threshold | Pets | CUB | Flowers | Birdsnap | Food | IN100 |
|---|---|---|---|---|---|---|
| 0 (**only real data**) | 93.6 / 0.40 | 83.1 / 0.13 | 87.4 / 0.40 | 73.0 / 0.00 | 86.8 / 0.63 | 87.4 / 0.20 |
| 1/5 training set | 94.4 / 0.56 | 83.6 / 0.25 | 90.0 / 0.40 | **73.6 / 0.00** | 87.4 / 0.63 | 89.1 / 0.40 |
| 1/4 training set | 94.2 / 0.48 | **83.9 / 0.25** | 89.9 / 0.50 | 73.5 / 0.00 | 87.5 / 0.63 | 89.6 / 0.40 |
| 1/2 training set | **94.6 / 0.56** | 83.6 / 0.25 | **90.1 / 0.50** | 73.4 / 0.00 | **87.8 / 0.64** | **89.9 / 0.40** |
| Full training set (Real-Fake) | 94.2 / 0.48 | 83.1 / 0.00 | 89.0 / 0.50 | 73.0 / 0.00 | 87.4 / 0.61 | 88.6 / 0.40 |

Table 10: Top-1 accuracy / worst-case disparity for different VoG filtering thresholds.

| VoG Filtering Threshold | Pets | CUB | Flowers | Birdsnap | Food | IN100 |
|---|---|---|---|---|---|---|
| 0% (Real-Fake) | 94.2 / 0.48 | 83.1 / 0.00 | 89.0 / 0.50 | 73.0 / 0.00 | 87.4 / 0.61 | 88.6 / 0.40 |
| 25% | 94.5 / 0.48 | 83.7 / 0.25 | 90.6 / 0.50 | 73.9 / 0.00 | **87.8 / 0.64** | **90.1 / 0.40** |
| 50% | 94.5 / 0.48 | **83.9 / 0.25** | 90.1 / 0.50 | 74.2 / 0.00 | 87.6 / 0.61 | 89.6 / 0.40 |
| 75% | **94.7 / 0.56** | 83.3 / 0.25 | **91.0 / 0.56** | **74.5 / 0.00** | 87.4 / 0.61 | 89.6 / 0.40 |
| 100% (**only real data**) | 93.6 / 0.40 | 83.1 / 0.13 | 87.4 / 0.40 | 73.0 / 0.00 | 86.8 / 0.63 | 87.4 / 0.20 |

**Impact of Synthetic Data Quantity.**    We examine the effect of varying real-to-synthetic data ratios, with additional experiments in Table 15.

- We ensure equivalent search space sizes across Real-Fake and GenDataAgent (e.g., Real-Fake 1:2 matches GenDataAgent 1:0.1). GenDataAgent consistently outperforms Real-Fake, particularly in larger search spaces.
- Real-Fake struggles to scale with synthetic data, with Real-Fake (1:10) underperforming Real-Fake (1:1) on fine-grained datasets like Pets and CUB. In contrast, GenDataAgent remains stable and excels at handling large synthetic datasets, leading to a growing performance gap over Real-Fake.

# G   ENHANCING EFFICIENCY

GenDataAgent (1:1) incurs higher training overhead than using only real data or the Real-Fake method. However, inference time remains unaffected across all methods, ensuring real-world deployment performance is not compromised.

To mitigate training overhead, we introduce two lightweight versions in our ablation study:

- **GenDataAgent (1:0.1):** Matches Real-Fake (1:1) in training time while achieving comparable or superior performance.
- **GenDataAgent (1:0.5):** Outperforms Real-Fake across all datasets with a moderate training cost increase.

Moreover, as shown in Figure 4 of the main paper, training time can be further reduced by decoupling specific components and utilizing more efficient SD and LLM variants. This highlights GenDataAgent's scalability and potential integration with future generative model optimizations.

To further improve efficiency, we propose two additional strategies:

- **Offline Mode:** SD adaptation, Llama-2 caption perturbations, and synthetic data generation are performed offline. The on-the-fly mechanism then selects relevant samples from a pre-generated pool, significantly reducing online training time. As shown in Figure 4, this reduces training primarily to classifier fine-tuning, which is comparatively fast.
- **Asynchronous Generation:** Synthetic data generation and classifier fine-tuning run in parallel, minimizing wait times by overlapping processes. This can further reduce overall training time and is an avenue for future research.

# H   IN-DISTRIBUTION ALIGNMENT AND OUT-OF-DISTRIBUTION GENERALIZATION

**In-Distribution Alignment.**    SD leverages the CLIP (Radford et al., 2021) text encoder for image generation. Maintaining consistent prompt formatting during LoRA fine-tuning and synthetic data

Table 11: Dataset statistics.

|  | Pets | CUB | Flowers | Birdsnap | Food | IN100 |
|---|---|---|---|---|---|---|
| Domain | Pet Breeds | Birds | Flowers | Birds | Food | Natural Images |
| # Training Samples | 3,680 | 5,994 | 2,040 | 47,386 | 75,750 | 126,689 |
| # Test Samples | 3,669 | 5,794 | 6,149 | 2,443 | 25,250 | 5,000 |
| # Classes | 37 | 200 | 102 | 500 | 101 | 100 |

Table 12: Training hyperparameters for downstream classification.

|  | Pets | CUB | Flowers | Birdsnap | Food | IN100 |
|---|---|---|---|---|---|---|
| On-the-fly Iterations | 20 | 20 | 20 | 20 | 20 | 20 |
| Train Res $\rightarrow$ Test Res | $224 \rightarrow 224$ | $448 \rightarrow 448$ | $224 \rightarrow 224$ | $224 \rightarrow 224$ | $224 \rightarrow 224$ | $224 \rightarrow 224$ |
| Epochs | 200 | 200 | 200 | 200 | 200 | 200 |
| Batch Size | $128 \times 8$ | $64 \times 8$ | $128 \times 8$ | $128 \times 8$ | $128 \times 8$ | $128 \times 8$ |
| Optimizer | SGD | SGD | SGD | SGD | SGD | SGD |
| Learning Rate | 0.1 | 0.2 | 0.1 | 0.1 | 0.1 | 0.1 |
| LR Decay | Multistep | Multistep | Multistep | Multistep | Multistep | Multistep |
| Decay Rate | 0.2 | 0.2 | 0.2 | 0.2 | 0.2 | 0.2 |
| Decay Epochs | 50/100/150 | 50/100/150 | 50/100/150 | 50/100/150 | 50/100/150 | 50/100/150 |
| Weight Decay | 5e-4 | 5e-4 | 5e-4 | 5e-4 | 5e-4 | 5e-4 |
| Mixed Precision | ✓ | ✓ | ✓ | ✓ | ✓ | ✓ |

generation stabilizes embeddings, reducing variance and improving alignment with the target dataset. While prompt consistency enhances embedding coherence, true alignment stems from LoRA fine-tuning, which refines SD's generative capabilities to match the target distribution.

**Out-of-Distribution Generalization.** While ensuring in-distribution alignment is crucial, incorporating outliers can improve generalization in out-of-distribution settings. As shown in Appendix D, stricter VoG filtering (fewer outliers) does not always yield the best results, suggesting that controlled outlier exposure enhances model robustness. For out-of-distribution scenarios, we recommend lowering the VoG filtering threshold to retain diverse and potentially out-of-distribution samples, strengthening adaptability to unseen data.

## I  ACTIVE LEARNING VS. MARGINAL SAMPLE SELECTION

**Active Learning:**

- **Objective:** Selects unlabeled samples for annotation based on informativeness, without prior label knowledge.
- **Behavior:** Uses static difficulty measures, independent of downstream model feedback.

**Marginal Sample Selection:**

- **Objective:** Utilizes synthetic sample labels to compute marginal scores, estimating difficulty relative to the downstream model.
- **Behavior:** Dynamically reassesses sample difficulty in an on-the-fly framework, adapting as the model evolves.
- **Focus:** Enhances synthetic training data, whereas active learning prioritizes real-world unlabeled data for annotation.

## J  INTEGRATION WITH DOMAIN ADAPTATION TECHNIQUES

Domain adaptation (DA) techniques (Tang & Jia, 2023) effectively mitigate distribution shifts between real and synthetic data, providing complementary insights to our approach.

While GenDataAgent aligns distributions via SD fine-tuning, it further enhances adaptation through:

- **Marginal Sample Selection:** Prioritizes challenging samples to improve model robustness.
- **LLM Caption Perturbation:** Enhances prompt diversity for richer synthetic data.
- **VoG Filtering:** Maintains in-distribution relevance by filtering outliers.

Table 13: Top-1 accuracy on clean and corrupted images (Gaussian Blur, Speckle Noise).

| Method | Pets | CUB | Flowers | Birdsnap | Food | IN100 |
|---|---|---|---|---|---|---|
| Real-Fake | 94.2 / 93.0 / 90.4 | 83.1 / 77.3 / 75.4 | 89.0 / 88.0 / 79.2 | 73.0 / 72.4 / 71.7 | 87.4 / 86.6 / 86.1 | 88.6 / 88.1 / 87.2 |
| GenDataAgent | **94.7 / 93.6 / 90.9** | **83.9 / 79.1 / 76.0** | **91.0 / 90.3 / 82.4** | **74.5 / 73.8 / 72.4** | **87.8 / 87.1 / 86.7** | **90.1 / 89.7 / 89.0** |

Table 14: Train accuracy, validation accuracy, and accuracy gap after convergence.

| Method | Pets | CUB | Flowers | Birdsnap | Food | IN100 |
|---|---|---|---|---|---|---|
| Real-Fake | 99.7 / 94.2 / 5.5 | 93.6 / 83.1 / 10.5 | 99.1 / 89.0 / 10.1 | 90.2 / 73.0 / 17.2 | 96.6 / 87.4 / 9.2 | 95.4 / 88.6 / 6.8 |
| GenDataAgent | 97.8 / 94.7 / 3.1 | 92.2 / 83.9 / 8.3 | 98.3 / 91.0 / 7.3 | 89.1 / 74.5 / 14.6 | 96.2 / 87.8 / 8.4 | 94.8 / 90.1 / 4.7 |

These techniques follow generalizable principles and can be seamlessly integrated with DA methods like those in Tang & Jia (2023).

## K GENERABILITY OF GENDATAAGENT

**Extension to Other TTI Models.** GenDataAgent is adaptable to other TTI generation models, though our experiments use SD v1.5 for consistency with prior work (Lei et al., 2023; Yuan et al., 2023). Models like SD v3 and Kandinsky differ in architecture, affecting the applicability of certain techniques:

- **Fine-Tuning Constraints:** SD v3 and Kandinsky lack official LoRA fine-tuning support for CLIP-based image features, making direct adaptation non-trivial.
- **Framework Adaptability:** Despite this, core principles like marginal sample selection, VoG filtering, and LLM-based caption perturbation remain generalizable and can be adapted with suitable modifications. We plan to explore these adaptations in future work.

**Applicability to Other CV Tasks.** While GenDataAgent is designed for classification tasks, its principles—marginal sample selection, LLM perturbation, and VoG filtering—are task-agnostic. Applying it to other CV tasks requires:

- **Task-Specific Modifications:** Adapting sampling and filtering techniques to align with different learning objectives.
- **Suitable Tasks:** Tasks reliant on paired data, such as object detection and semantic segmentation, are strong candidates.

The literature (Yuan et al., 2023; Lei et al., 2023; Sarıyıldız et al., 2023; Li et al., 2023a; He et al., 2022) has primarily focused on classification, necessitating methodological adaptations for fair evaluation in other tasks. We leave this for future work given the additional complexity involved.

## L LIMITATIONS

**Dependence on Generative Models.** The effectiveness of GenDataAgent relies on stable diffusion's ability to generate high-quality synthetic data. Performance may vary across different models and versions, affecting downstream classification.

**Efficiency of Synthetic Data Generation.** Generating high-quality synthetic data is a key bottleneck in augmentation. While techniques like Bolya & Hoffman (2023) improve efficiency, generation remains costly, and output quality can degrade.

**Applicability Beyond Classification.** Currently, GenDataAgent is tailored for classification. Expanding to other tasks (e.g., object detection, segmentation) may require adapting stable diffusion and sampling strategies.

**Automating Threshold Selection.** GenDataAgent uses manually set thresholds. Future work could explore adaptive methods for more automated and data-driven threshold tuning.

Table 15: Ablation study on synthetic data quantity.

| Method | Pets | CUB | Flowers |
|---|---|---|---|
| Only real data | 93.6 | 83.1 | 87.4 |
| Real-Fake (1:1) | 94.2 | 83.1 | 89.0 |
| Real-Fake (1:2) | 93.6 | 82.4 | 88.8 |
| GenDataAgent (1:0.1) | 94.3 | 84.1 | 90.2 |
| Real-Fake (1:5) | 93.3 | 80.7 | 88.3 |
| GenDataAgent (1:0.25) | 94.4 | 84.2 | 91.4 |
| Real-Fake (1:10) | 93.3 | 78.4 | 87.7 |
| GenDataAgent (1:0.5) | 94.4 | 84.5 | 91.5 |
| GenDataAgent (1:1) | **94.7** | **83.9** | **91.0** |

