# OpenReview forum: "GenDataAgent: On-the-fly Dataset Augmentation with Synthetic Data"
_ICLR.cc/2025/Conference — ICLR 2025 Poster_

### Official Review · Reviewer_NMJi · 2024-10-29

**Soundness:** 3
**Presentation:** 3
**Contribution:** 2
**Rating:** 6
**Confidence:** 4

**Summary:**

This paper investigates how to leverage synthetic data generated by text-to-image generation models, such as Stable Diffusion (SD), to enhance image classification performance in terms of both accuracy and fairness. The authors propose a sampling strategy that prioritizes sample utility, which is determined by the proximity of a sample to the decision boundary, as well as in-distributional data samples, using the VoG score. By integrating a series of reasonable treatment, the proposed method achieves state-of-the-art performance across different image classification benchmarks.

**Strengths:**

The paper is well-written, providing clear motivations and explanations of the proposed method. The introduced metric for valuing synthetic data samples, based on their utility and proximity to the decision boundary, is reasonable. The integration of different techniques, such as perturbing text prompts to enhance diversity and ensuring alignment with the target distribution, appears well-justified and appropriate for the stated problem. The proposed method also demonstrates state-of-the-art performance across all kinds of image classification benchmarks.

**Weaknesses:**

The paper's critical weakness lies in its lack of novelty, as most of the techniques employed, including the VoG metric, are derived from previous works. This aspect positions the paper more as an engineering-heavy work rather than an original contribution. Specifically, the extensively discussed VoG metric appears somewhat subjective, as the choice of using gradients from epochs 10, 20, and 30 to compute the metric lacks proper justification or guidelines for other benchmarks or tasks. Moreover, while the proposed method demonstrates significant improvement over using only real data, it performs only marginally better than other synthetic augmentation methods, such as Real-Fake and Internet Explorer, while also taking much longer GPU hours. In fact, synthetic data augmentation has been extensively studied in the context of image classification, and the observed improvement may merely be a result of overfitting to those small-scale benchmarks and may not generalize to real-world or in-the-wild settings.

**Questions:**

I'm willing to raise the score if the following questions are well-addressed:

1. Can you also include robustness metrics, such as accuracy under common corruptions and perturbations as well as adversarial settings? Is the proposed method still better than the baselines?
2. Would the proposed method generalize across different Text2Image generation methods, such as SD3 and Kandinsky (does not have to be the two)?
3. Would the proposed method generalize to other discriminative CV tasks such as image retrieval or object detection?
4. Line 474, the statement that accuracy is highly correlated with the synthetic data volume seems to be contradictory with a concurrent ICLR25 submission (Mousterian: exploring the equivalence of generative and real data augmentation in classification). In that work, they state that synthetic data is more valuable when the quantity is small, yet its value diminishes quickly rather than staying correlated with accuracy.
5. The discussion on overfitting in line 483 requires more clarity and experimental evidence to support the strong statement made. The current evidence on Pets dataset appears quite weak and insufficient.

---

> ### Author Response · Authors · 2024-11-26
> **Response to Reviewer NMJi (part 1)**
>
> `Q-1:` The paper's critical weakness lies in its lack of novelty, as most of the techniques employed, including the VoG metric, are derived from previous works. This aspect positions the paper more as an engineering-heavy work rather than an original contribution.
>
> `A-1:` Leveraging large volumes of synthetic data for downstream tasks remains a significant challenge, with **no comprehensive pipeline in existing literature**. GenDataAgent addresses this gap by introducing a unified framework that ensures synthetic data is diverse, in-distribution, and task-specific through:
> - LLM-based Caption Perturbation: Repurposes LLMs, traditionally used for text generation, to diversify image captions for enhanced synthetic data variation.
> - VoG Filtering: Adapts the VoG metric from real datasets to detect outliers in synthetic data, aligning it with the target distribution.
> - Marginal Sample Utilization: Instead of applying weighted losses, uses challenging samples to guide synthetic data generation, dynamically aligning with model requirements.
>
> Though GenDataAgent involves engineering aspects, its principled framework lays a foundation for future advancements by integrating and repurposing existing techniques into a cohesive pipeline.
>
> `Q-2:` Specifically, the extensively discussed VoG metric appears somewhat subjective, as the choice of using gradients from epochs 10, 20, and 30 to compute the metric lacks proper justification or guidelines for other benchmarks or tasks.
>
> `A-2:` The choice of epochs 10, 20, and 30 for computing the VoG metric corresponds to the 1st, 2nd, and 3rd iterations of on-the-fly fine-tuning, balancing model refinement with resource efficiency. Our ablation study (Table 5) shows that increasing the number of checkpoints does not significantly improve performance. Therefore, we use the minimum of 3 checkpoints to reduce computational overhead while maintaining effectiveness.
>
> `Q-3:` Moreover, while the proposed method demonstrates significant improvement over using only real data, it performs only marginally better than other synthetic augmentation methods, such as Real-Fake and Internet Explorer, while also taking much longer GPU hours.
>
> `A-3:`
> 1. Our GenDataAgent is developed upon the SOTA method Real-Fake, which serves as **a strong baseline with saturate accuracy** in the "synthetic data augmentation" setup. Furthermore, the proposed techniques demonstrate **greater improvements in the "synthetic data only" setting**, as illustrated in Table 1. Additionally, Internet Explorer uses part of our components (SD adaptation, LLaMA caption perturbation) so that it also performs well.
> 2. In our ablation study, we also introduce **two lightweight versions**: GenDataAgent (1:0.1) and GenDataAgent (1:0.5). As illustrated in Figure 4, the training time for GenDataAgent (1:0.1) is close to that of Real-Fake (1:1), while delivering comparable or superior performance. GenDataAgent (1:0.5) already exceeds the SOTA Real-Fake across all datasets. Additionally, Figure 4 shows a breakdown of training time, indicating that further acceleration can be achieved by using more efficient SD and LLM models.
>
> `Q-4:` In fact, synthetic data augmentation has been extensively studied in the context of image classification, and the observed improvement may merely be a result of overfitting to those small-scale benchmarks and may not generalize to real-world or in-the-wild settings.
>
> `A-4:` Our focus on small-scale benchmarks is intended to demonstrate that synthetic data can address gaps in training data by covering cases that might otherwise be ignored. Importantly, the results suggest that synthetic data augmentation is unlikely to lead to overfitting on these benchmarks:
>
> - If synthetic data augmentation caused overfitting to small-scale benchmarks, increasing the volume of synthetic data would likely continue to improve performance. However, as shown in Figure 4, performance drops when using 10x synthetic data (Real-Fake 1:10). This indicates that overfitting is not a significant concern in this context.
> - The GenDataAgent pipeline provides flexibility by allowing the user to adjust the VoG filtering threshold. This balances diversity with in-distribution alignment, ensuring that the synthetic data contributes meaningfully without overwhelming the training process. This adaptability makes GenDataAgent more robust than static augmentation methods like Real-Fake, which lack dynamic controls.

---

> ### Author Response · Authors · 2024-11-26
> **Response to Reviewer NMJi (part 2)**
>
> `Q-5:` Can you also include robustness metrics, such as accuracy under common corruptions and perturbations as well as adversarial settings? Is the proposed method still better than the baselines?
>
> `A-5:` To evaluate robustness, we introduced common corruptions such as **Gaussian Blur and Speckle Noise**, following the methodology outlined in [1]. All methods were trained on clean images and tested on corrupted images to assess their robustness.
>
> Below is a comparison of our GenDataAgent with the SOTA baseline Real-Fake across six datasets:
>
> | **Clean / Gaussian Blur / Speckle Noise** | **Pets**         | **CUB**          | **Flowers**      | **Birdsnap**     | **Food**         | **IN100**        |
> |-|-|-|-|-|-|-|
> | RealFake                              | 94.2 / 93.0 / 90.4 | 83.1 / 77.3 / 75.4 | 89.0 / 88.0 / 79.2 | 73.0 / 72.4 / 71.7 | 87.4 / 86.6 / 86.1 | 88.6 / 88.1 / 87.2 |
> | GenDataAgent                          | 94.7 / 93.6 / 90.9 | 83.9 / 79.1 / 76.0 | 91.0 / 90.3 / 82.4 | 74.5 / 73.8 / 72.4 | 87.8 / 87.1 / 86.7 | 90.1 / 89.7 / 89.0 |
>
>
> Our method consistently **outperforms Real-Fake across all datasets and corruptions**. We will include these results in the final version of the manuscript.
>
> [1] Benchmarking Neural Network Robustness to Common Corruptions and Perturbations. ICLR, 2019.
>
> `Q-6:` Would the proposed method generalize across different Text2Image generation methods, such as SD3 and Kandinsky (does not have to be the two)?
>
> `A-6:` The proposed GenDataAgent framework is indeed **extendable to other TTI generation methods**. However, the choice of SD1.5 in our experiments was motivated by its status as a common baseline in the literature [2,3], ensuring a fair and consistent comparison with existing methods.
>
> It is important to note that SD3 and Kandinsky employ different architectures and mechanisms, which could affect the application of certain techniques used in GenDataAgent:
>
> - Fine-Tuning Challenges: Both SD3 and Kandinsky lack official support for LoRA fine-tuning with CLIP-based image features, making it non-trivial to directly adapt these models to the target dataset.
> - Framework Adaptability: Despite these challenges, the principles of GenDataAgent—such as marginal sample selection, VoG filtering, and LLM-based caption perturbation—are general and can be adapted to other Text-to-Image models with suitable modifications.
> We plan to explore these adaptations in future work and will include a detailed discussion on the potential challenges and solutions for extending GenDataAgent to different TTI models in the final version.
>
> [2] Image Captions are Natural Prompts for Text-to-Image Models, arxiv, 2023.
>
> [3] REAL-FAKE: EFFECTIVE TRAINING DATA SYNTHESIS THROUGH DISTRIBUTION MATCHING, ICLR, 2024.
>
> `Q-7:` Would the proposed method generalize to other discriminative CV tasks such as image retrieval or object detection?
>
> `A-7:` Our **GenDataAgent can be extended to other computer vision tasks** involving paired data, such as object detection and image segmentation. For instance, in object detection tasks, given bounding boxes and categories, generative models like GLIGEN [4] and LayoutDM [5] can be employed to synthesize corresponding images. Similarly, given a segmentation map, UniControl [6] can generate the corresponding images. In other words, as long as paired data exists, GenDataAgent can be adapted to these tasks by substituting the general Stable Diffusion model with task-specific controllable generative models.
>
> [4] GLIGEN: Open-Set Grounded Text-to-Image Generation, CVPR, 2023.
>
> [5] LayoutDiffusion: Controllable Diffusion Model for Layout-to-Image Generation, CVPR, 2023.
>
> [6] UniControl: A Unified Diffusion Model for Controllable Visual Generation in the Wild, NeurIPS, 2023.

---

> ### Author Response · Authors · 2024-11-26
> **Response to Reviewer NMJi (part 3)**
>
> `Q-8:` Line 474, the statement that accuracy is highly correlated with the synthetic data volume seems to be contradictory with a concurrent ICLR25 submission (Mousterian: exploring the equivalence of generative and real data augmentation in classification). In that work, they state that synthetic data is more valuable when the quantity is small, yet its value diminishes quickly rather than staying correlated with accuracy.
>
> `A-8:`
> 1. These are two conclusions derived under different conditions.
> - Mousterian claimed that the accuracy improvement from incorporating synthetic data depends on the volume of real data. As stated in Line 350, “Integrating synthetic data greatly enhances classification performance when real data is scarce, but the benefit decreases as real data becomes more plentiful.”
> - In contrast, we argued that, given the same amount of real data, the performance gain for each category is correlated with the volume of synthetic data. However, this correlation does not imply proportionality.
> 2. That said, we share part of the conclusion that the improvement is not proportional to the volume of synthetic data (as shown in Figure 4). However, within a certain threshold, an increase in synthetic data volume is correlated with accuracy improvement (Figure 5).
>
> By the way, they have withdrawn their paper for some reasons.
>
> `Q-9:` The discussion on overfitting in line 483 requires more clarity and experimental evidence to support the strong statement made. The current evidence on Pets dataset appears quite weak and insufficient.
>
> `A-9:` Thank you for your suggestion. Below, we present the Train Accuracy, Validation Accuracy, and Accuracy Gap after convergence for all datasets. Notably, the train-validation accuracy gap is reduced on GenDataAgent, which can be seen as a form of mitigation for overfitting. The reason might be the synthetic data covers some cases that the real data ignore (the diversity introduced by LLaMA caption perturbation), thus enhancing the model's robustness and generalization capabilities.
>
> | **Train Acc / Val Acc / Acc Gap**     | **Pets**        | **CUB**         | **Flowers**     | **Birdsnap**    | **Food**       | **IN100**      |
> |---------------------------------------|-----------------|-----------------|-----------------|-----------------|----------------|----------------|
> | RealFake                              | 99.7 / 94.2 / 5.5 | 93.6 / 83.1 / 10.5 | 99.1 / 89.0 / 10.1 | 90.2 / 73.0 / 17.2 | 96.6 / 87.4 / 9.2  | 95.4 / 88.6 / 6.8  |
> | GenDataAgent (Ours)                   | 97.8 / 94.7 / 3.1 | 92.2 / 83.9 / 8.3  | 98.3 / 91.0 / 7.3  | 89.1 / 74.5 / 14.6 | 96.2 / 87.8 / 8.4  | 94.8 / 90.1 / 4.7  |

---

> > ### Comment · Reviewer_NMJi · 2024-11-26
> >
> > Thank you for your detailed response. I appreciate the ablation studies and the thorough explanation. However, several of my questions remain unanswered. For instance, the method is described as a principled framework that lays the foundation for future advancements—a bold claim—but there are no experiments demonstrating its applicability to tasks like object detection or other computer vision problems. Additionally, some claims would benefit from greater clarity and more robust data-driven validation. For example, an ablation study evaluating how "synthetic data quantity correlates with accuracy improvement" would be a straightforward and valuable experiment to include. Given these concerns, I believe it is reasonable to maintain the current score.

---

> > > ### Author Response · Authors · 2024-11-28
> > >
> > > Dear, Reviewer NMJi
> > >
> > > Thank you for your valuable feedback and for taking the time to review our response. Below, we provide additional explanations and experiments, which we hope will help clarify any remaining concerns.
> > >
> > > **1. Generability of our method**
> > > - In our previous response, we aimed to clarify the potential of our method for adapting to other CV tasks, in response to your insightful inquiry. However, we did not intend to imply that our method was designed for general applications in the manuscript. Instead, we present it specifically within the context of classification tasks, as these are among the most widely studied areas in the field [1,2,3,4,5]. We will ensure that this distinction is made clearer in the final version.
> > > - We truly appreciate your suggestions regarding the extension of our approach to other tasks and the in-depth analysis of its potential. Here are our responses:
> > >     - The principles behind our method (including marginal sample selection, LLM perturbation diversification, and in-distribution VoG filtering) are not task-specific, making them theoretically applicable to a broader range of tasks.
> > >     - Since the downstream models and baseline methods are different, appropriate modifications are necessary for this type of extension.
> > >     - Furthermore, we clarify the tasks where our framework might be applicable. Specifically, since synthetic data generation relies on paired data, tasks such as object detection and semantic segmentation are suitable applications.
> > >
> > > - Since the community has primarily focused on classification tasks [1,2,3,4,5], conducting further experiments on other tasks will require modifications to the proposed method as well as adjustments to the comparison methods. Given the complexities involved—such as differences in downstream models, baseline approaches, and the need for a fair evaluation—these experiments are scheduled for future work due to the additional time and effort required.
> > >
> > > [1] Fake it till you make it: Learning transferable representations from synthetic imagenet clones. CVPR, 2023.
> > >
> > > [2] Image captions are natural prompts for text-to-image models. arxiv, 2023.
> > >
> > > [3] Is synthetic data from generative models ready for image recognition? ICLR, 2023.
> > >
> > > [4] Real-fake: Effective training data synthesis through distribution matching. ICLR, 2024.
> > >
> > > [5] Internet explorer: Targeted representation learning on the open web. ICML, 2023.
> > >
> > > **2. Ablation study on synthetic data quantity**
> > > - Thank you for the suggestion. We illustrated how synthetic data quantity impacts accuracy in Figure 4 in our manuscript. Below, we further organized the data into a table for easier understanding.
> > >
> > > | Methods                      | Pets  | CUB  | Flowers | Birdsnap | Food  | IN100 |
> > > |-|-|-|-|-|-|-|
> > > | Only Real                    | 93.6  | 83.1 | 87.4    | 73.0     | 86.8  | 87.4  |
> > > | Real-Fake (1:1)              | 94.2  | 83.1 | 89.0    | 73.0     | 87.4  | 88.6  |
> > > | Real-Fake (1:10)             | 93.3  | 78.4 | 87.7    | 67.4     | 86.5  | 89.1  |
> > > | GenDataAgent (1:0.1)         | 94.3  | 84.1 | 90.2    | 74.4     | 87.4  | 88.6  |
> > > | GenDataAgent (1:0.5)         | 94.4  | 84.5 | 91.5    | 74.4     | 87.7  | 89.5  |
> > > | GenDataAgent (1:1)           | 94.7  | 83.9 | 91.0    | 74.5     | 87.8  | 90.1  |
> > >
> > > - Besides, we conduct additional experiments with a wider range of real-to-synthetic ratios. Due to time and resource constraints, these experiments are conducted on Pets, CUB, and Flowers datasets.
> > >
> > > | Methods                      | Pets  | CUB  | Flowers |
> > > |-|-|-|-|
> > > | Only Real                    | 93.6  | 83.1 | 87.4    |
> > > | Real-Fake (1:1)              | 94.2  | 83.1 | 89.0    |
> > > |-|-|-|-|
> > > | Real-Fake (1:2)              | 93.6  | 82.4 | 88.8    |
> > > | GenDataAgent (1:0.1)         | 94.3  | 84.1 | 90.2    |
> > > |-|-|-|-|
> > > | Real-Fake (1:5)              | 93.3  | 80.7 | 88.3    |
> > > | GenDataAgent (1:0.25)        | 94.4  | 84.2 | 91.4    |
> > > |-|-|-|-|
> > > | Real-Fake (1:10)             | 93.3  | 78.4 | 87.7    |
> > > | GenDataAgent (1:0.5)         | 94.4  | 84.5 | 91.5    |
> > > |-|-|-|-|
> > > | GenDataAgent (1:1)           | 94.7  | 83.9 | 91.0    |
> > >
> > > - To ensure a fair comparison, we evaluate both Real-Fake and our GenDataAgent within the same search space size (for example, Real-Fake 1:2 shares the same search space size as GenDataAgent 1:0.1). In all cases, GenDataAgent consistently outperforms Real-Fake, especially in larger search spaces.
> > > - Notably, Real-Fake struggles to scale with increasing synthetic data, with Real-Fake (1:10) performing even worse than using only real data on the Pets and CUB datasets. In contrast, GenDataAgent maintains stable performance across all real-to-synthetic ratios and excels when handling large quantities of synthetic data, leading to a significantly larger performance gap over Real-Fake.
> > >
> > > We hope these additional explanations and experiments further address your concerns and assist you in making your final decision.
> > >
> > > Sincerely,
> > >
> > > Authors

---

> > > > ### Comment · Reviewer_NMJi · 2024-11-28
> > > >
> > > > I’m satisfied with the author’s final response, which I find careful and accurate. Trusting that the authors will incorporate these final responses into the paper, I am raising my score to a 6.

---

> > > > > ### Author Response · Authors · 2024-11-28
> > > > >
> > > > > Dear, Reviewer NMJi
> > > > >
> > > > > Thank you for your continued feedback and for raising the score! We promise to incorporate these final responses into the final version.
> > > > >
> > > > > Sincerely,
> > > > >
> > > > > Authors

---

### Official Review · Reviewer_rpaL · 2024-11-03

**Soundness:** 2
**Presentation:** 3
**Contribution:** 3
**Rating:** 8
**Confidence:** 4

**Summary:**

The paper presents a novel approach for training dataset augmentation in computer vision using a generative agent, GenDataAgent. This method addresses the limitations of previous works that uniformly search across the category space and fail to consider the interaction between synthetic data generation and downstream task training. The proposed agent generates high-quality synthetic data on-the-fly, ensuring alignment with the target training dataset distribution while prioritizing diversity and relevance. Evaluations demonstrate that this approach enhances the generalization performance of downstream models fine-tuned on the augmented datasets.

**Strengths:**

- The introduction of GenDataAgent for on-the-fly synthetic data generation represents a significant advancement in training dataset augmentation, addressing the critical challenges of distribution alignment and sample diversity.

- The method effectively enhances the generalization performance of downstream models by prioritizing the sampling of diverse synthetic data that complements marginal training samples, which is crucial for improving model robustness.

- Unlike prior research, this work does not rely on specific distributional assumptions, making the approach more versatile and applicable to a wider range of datasets, including those with varying class distributions and limited examples.

**Weaknesses:**

- There exists a conflict: tailoring synthetic data generation for specific downstream tasks limits its generality, while using it for general purposes diminishes its specialty. The authors could provide further insights to clarify this issue.

- Is it necessary for synthetic data to be in-distribution? While aligning the distribution of synthetic data with the training data is important, it does not align with the distribution of the test data, particularly in out-of-distribution (OOD) scenarios. This raises the question of whether this paper focuses solely on generalization within in-distribution contexts, potentially limiting its applicability to real-world situations where test data may vary significantly. The authors may evaluate the proposed agent's capability for OOD generalization.

- Though the on-the-fly generation of relevant data saves the storage cost and improves relevance to the task at hand, how reusable and sustainable is synthetic data?

- For marginal training samples, synthetic data that exhibit higher variance in gradient updates are specifically chosen. Does this imply that these synthetic samples are those that the current model struggles to learn effectively? If so, it suggests that these samples may be more challenging or informative, potentially indicating areas where the model's performance is weak. This focus on high-variance samples could enhance the model's robustness by addressing its limitations and improving its ability to generalize to diverse data distributions. Further clarification on this selection strategy and its implications for model training would be beneficial, e.g., discussing it with the selection criteria in active learning.

- This paper would benefit from a more comprehensive literature review. Specifically, it overlooks some closely related works, such as [a]. I recommend that the authors include a discussion of this work and its connections to their research. This will help to contextualize their contributions within the existing body of knowledge.

[a] A New Benchmark: On the Utility of Synthetic Data with Blender for Bare Supervised Learning and Downstream Domain Adaptation. CVPR, 2023.

- The authors claim that maintaining a consistent prompt format for both adapting stable diffusion and generating synthetic data ensures greater alignment between the distribution of the generated data and the target data. However, it is not entirely clear how a consistent prompt format alone guarantees this distribution alignment. I recommend that the authors provide a more detailed explanation of this relationship.

- Additionally, it would be valuable to discuss whether incorporating domain adaptation (DA) techniques, as described in [a], during the model fine-tuning process could lead to further performance improvements. Exploring this potential synergy could enhance the robustness of the findings presented in the manuscript.

**Questions:**

See Weaknesses.

---

> ### Author Response · Authors · 2024-11-26
> **Response to Reviewer rpaL (part 1)**
>
> `Q-1:` There exists a conflict: tailoring synthetic data generation for specific downstream tasks limits its generality, while using it for general purposes diminishes its specialty. The authors could provide further insights to clarify this issue.
>
> `A-1:` Adapting SD to a target dataset via LoRA fine-tuning tailors synthetic data generation to specific tasks, potentially reducing generalizability to other datasets. However, this does not imply overfitting, as SD’s inherent diversity ensures coverage of cases missing in limited real data.
>
> **GenDataAgent offers flexibility to balance this trade-off**:
> - For Robustness: Raising the VoG threshold accepts more outliers, increasing diversity and enhancing generalization.
> - For Specificity: Lowering the VoG threshold focuses on a specific region, optimizing task-specific performance.
> This balance between generality and specificity is a design choice, and GenDataAgent provides tools to adapt as needed.
>
> `Q-2:` Is it necessary for synthetic data to be in-distribution? While aligning the distribution of synthetic data with the training data is important, it does not align with the distribution of the test data, particularly in out-of-distribution (OOD) scenarios. This raises the question of whether this paper focuses solely on generalization within in-distribution contexts, potentially limiting its applicability to real-world situations where test data may vary significantly. The authors may evaluate the proposed agent's capability for OOD generalization.
>
> `A-2:` Thank you for the observation. While keeping synthetic data in-distribution is important, including outliers can improve generalization, especially in OOD scenarios.
>
> - As shown in Table 4 (Appendix D), higher VoG filtering ratios (fewer outliers) don’t always yield the best results, suggesting that outliers can expose the model to valuable variations.
> - For OOD scenarios, we recommend lowering the VoG filtering threshold to include more diverse and potentially OOD samples, enhancing robustness to unseen data.
>
> We will expand on OOD generalization in the final version.
>
> `Q-3:` Though the on-the-fly generation of relevant data saves the storage cost and improves relevance to the task at hand, how reusable and sustainable is synthetic data?
>
> `A-3:`
> 1. We also provide an **offline version of GenDataAgent**, where the synthetic data generation is done before training and on-the-fly generation becomes on-the-fly selection. In this way, all synthetic data are saved and thus reusable. That’s a trade-off between storage efficiency and synthetic data reusable.
> 2. Moreover, an ablation study for threshold selection (for instance, adjusting VoG filtering threshold for different scenarios) can be realized as all synthetic data are saved.
>
> `Q-4:` For marginal training samples, synthetic data that exhibit higher variance in gradient updates are specifically chosen. Does this imply that these synthetic samples are those that the current model struggles to learn effectively? If so, it suggests that these samples may be more challenging or informative, potentially indicating areas where the model's performance is weak. This focus on high-variance samples could enhance the model's robustness by addressing its limitations and improving its ability to generalize to diverse data distributions. Further clarification on this selection strategy and its implications for model training would be beneficial, e.g., discussing it with the selection criteria in active learning.
>
> `A-4:` Yes, the marginal sample selection strategy targets synthetic samples with high variance in gradient updates, indicating areas where the model struggles. This approach enhances robustness and generalization by focusing on challenging regions.
>
> Active Learning:
> - **Objective:** Selects unlabeled samples for labeling, focusing on informativeness without prior label knowledge.
> - **Dynamic Behavior:** Relies on static difficulty measurements without feedback from the downstream task.
>
> Marginal Sample Selection:
> - **Objective:** Leverages synthetic sample labels to compute marginal scores, estimating difficulty relative to the downstream model.
> - **Dynamic Behavior:** Dynamically reassesses sample difficulty in an on-the-fly framework as the downstream model evolves, enabling adaptive selection.
> - **Focus:** Marginal sample selection specifically enhances synthetic data for training, whereas active learning targets real-world unlabeled data for annotation.
>
> We will include this discussion in the final paper to clarify these distinctions and their implications.

---

> ### Author Response · Authors · 2024-11-26
> **Response to Reviewer rpaL (part 2)**
>
> `Q-5:` This paper would benefit from a more comprehensive literature review. Specifically, it overlooks some closely related works, such as [a]. I recommend that the authors include a discussion of this work and its connections to their research. This will help to contextualize their contributions within the existing body of knowledge.
> [a] A New Benchmark: On the Utility of Synthetic Data with Blender for Bare Supervised Learning and Downstream Domain Adaptation. CVPR, 2023.
>
> `A-5:` Thank you for the valuable suggestion. Incorporating [a] will strengthen the contextualization of our work.
>
> 1. The domain adaptation (DA) techniques in [a] effectively reduce distribution shifts between real and synthetic data, **offering insights that complement our approach**.
>
> 2. GenDataAgent also aligns distributions through SD fine-tuning, it adds:
> - Marginal Sample Selection: Generates challenging samples to address model weaknesses.
> - LLM Caption Perturbation: Increases prompt diversity for richer synthetic data.
> - VoG Filtering: Ensures synthetic data remains in-distribution and relevant.
>
> These techniques are designed by general principle and can be combined with other techniques like DA in [a]. We **have incorporated [a] into the related work section**, highlighted in red, and will provide a more detailed discussion in the final version.
>
> `Q-6:` The authors claim that maintaining a consistent prompt format for both adapting stable diffusion and generating synthetic data ensures greater alignment between the distribution of the generated data and the target data. However, it is not entirely clear how a consistent prompt format alone guarantees this distribution alignment. I recommend that the authors provide a more detailed explanation of this relationship.
>
> `A-6:` Thank you for the feedback. SD uses the CLIP [1] text encoder to generate embeddings for image generation. Consistent prompt formatting during LoRA fine-tuning and synthetic data generation ensures stable embeddings, reducing variance and improving alignment with the target dataset.
>
> However, distribution alignment relies more on LoRA fine-tuning, which adapts SD to the target dataset by refining its generative capabilities. Consistent prompts enhance embedding coherence, but fine-tuning ensures alignment by tailoring the model to the domain.
>
> We will clarify these complementary roles in the final version.
>
> [1] Contrastive Language-Image Pre-Training, ICML, 2021.
>
> `Q-7:` Additionally, it would be valuable to discuss whether incorporating domain adaptation (DA) techniques, as described in [a], during the model fine-tuning process could lead to further performance improvements. Exploring this potential synergy could enhance the robustness of the findings presented in the manuscript.
>
> `A-7:` Thank you for the insightful suggestion. Incorporating domain adaptation techniques from [a] during the model fine-tuning process could indeed further enhance the alignment between the generated synthetic data and the target dataset distribution. This alignment has the potential to improve the quality of synthetic data, addressing one of the key bottlenecks in synthetic data augmentation. We will explore this integration and discuss its potential impact in the final version of the manuscript.

---

> ### Comment · Reviewer_rpaL · 2024-11-26
>
> I have reviewed the authors' rebuttal along with the comments from the other reviewers. I appreciate the thorough discussions and analyses provided, which effectively address many of my concerns.
>
> The revised paper adequately incorporates the suggested works for discussion, which is commendable. For the final version, I expect the authors to include the valuable analyses and discussions from the rebuttal in the main paper or supplemental material.
>
> Given these improvements, I will raise my current rating.

---

> > ### Author Response · Authors · 2024-11-28
> >
> > Dear, Reviewer rpaL
> >
> > Thank you for raising the rating. We are glad to know that our responses addressed your concerns, and we assure you that the analysis and discussion will be included in the final version.
> >
> > Sincerely,
> >
> > Authors

---

### Official Review · Reviewer_rGcL · 2024-11-03

**Soundness:** 4
**Presentation:** 4
**Contribution:** 3
**Rating:** 8
**Confidence:** 3

**Summary:**

The paper introduces the on-the-fly data augmentation method. The method first selects important samples using a marginal score. It then performs caption perturbations to increase the diversity of the data. Finally, outliers are filtered out using the gradients at the initial training iterations. The experiments show that the method effectively improves the accuracy.

**Strengths:**

- The paper is easy to follow.
- The paper includes clear motivation and a detailed analysis of the proposed framework.
- The experiments show the benefit of the proposed on-the-fly data augmentation method.

**Weaknesses:**

The proposed method is slow --- it can be up to 350 times slower than using only real data and 2 times slower than the comparing method (Real-Fake). Although the accuracy increases, this can be a major bottleneck.

**Questions:**

Related to the speed issue, I assume the reason is because it's the on-the-fly method. Can the method be adapted/switched between online and offline modes based on its need? Are there other ways to improve the speed?

---

> ### Author Response · Authors · 2024-11-26
> **Response to Reviewer rGcL**
>
> `Q-1:` The proposed method is slow --- it can be up to 350 times slower than using only real data and 2 times slower than the comparing method (Real-Fake). Although the accuracy increases, this can be a major bottleneck.
>
> `A-1:` While the training overhead of GenDataAgent (1:1) is higher compared to using only real data or the Real-Fake method, it is important to note that **inference time remains unaffected across all methods**. This ensures that real-world application performance, where inference time is critical, is not impacted by the additional training cost.
>
> To address the training overhead, we provide **two lightweight versions** in the ablation study:
> - GenDataAgent (1:0.1): Achieves comparable training time to Real-Fake (1:1) while delivering performance that is on par or better.
> - GenDataAgent (1:0.5): Surpasses the performance of Real-Fake across all datasets with a moderate increase in training cost.
>
> Furthermore, as shown in Figure 4, the training time can be further optimized by decoupling specific components and employing more efficient versions of SD and LLMs. This demonstrates the scalability of GenDataAgent and its potential for integration with future advancements in generative model efficiency.
>
> `Q-2:` Related to the speed issue, I assume the reason is because it's the on-the-fly method. Can the method be adapted/switched between online and offline modes based on its need? Are there other ways to improve the speed?
>
> `A-2:` Yes, **GenDataAgent can be adapted to include an offline mode** for situations where speed is critical. Specifically:
> - Offline Mode: Key processes such as SD adaptation, LLaMA caption perturbation, and synthetic data generation can be performed offline. The on-the-fly mechanism then shifts to selecting relevant samples from a pre-generated synthetic data pool, significantly reducing online training time. As shown in Figure 4, this reduces the training phase to primarily classifier fine-tuning, which is comparatively fast.
>
> Additionally, there are potential ways to further accelerate the process:
> - Asynchronous Generation: Synthetic data generation and classifier fine-tuning could be parallelized. This approach minimizes waiting times by overlapping the generation and training processes, reducing the overall duration. This can be explored in future work.

---

> > ### Comment · Reviewer_rGcL · 2024-11-26
> >
> > Thank you for your answer.
> > I ask the authors to include this discussion in the final version. This is an important one.
> > I maintain my positive score.

---

> > > ### Author Response · Authors · 2024-11-28
> > >
> > > Dear, Reviewer rGcL
> > >
> > > Thank you for your response. We are delighted to see that our answers were able to address your concerns, and we promise to include this discussion in the final version.
> > >
> > > Sincerely,
> > >
> > > Authors

---

### Official Review · Reviewer_Ln52 · 2024-11-06

**Soundness:** 2
**Presentation:** 3
**Contribution:** 1
**Rating:** 3
**Confidence:** 4

**Summary:**

The paper presents an algorithm for selecting synthetic data generated from Stable Diffusion, adjusting their contribution, and modifying the generation diversity for better closed-set classification model training as the downstream application. The main hypothesis is around the introduction of downstream training signal to provide feedback to the model generation process to better guide and use generated samples for discriminative model training.

**Strengths:**

The intuition of guiding the sample generation process from signals or feedback from the downstream model training process makes sense. The paper has presented results of both training completely with synthetic data and using synthetic data as augmentations. And they show the proposed method outperforms baselines compared.

**Weaknesses:**

Lack of technical novelty. Though the motivation of the idea makes sense, my biggest concern is on the technical novelty of the proposed algorithm. The proposed design components on ensuring sample diversity, generation distribution alignment with real data distribution, and the marginal sample selection are all coming from existing literature or prior works.

Lack of sufficient experiment results to validate the effectiveness of the proposed methods and the claims in the contributions. The contribution claims the effectiveness of the proposed method from the three aspects of marginal sample selection, sample diversity and the distribution alignment to real data distribution. However, from the ablation study, it appears that even without these proposed improvements, the classification model trained with synthetic data also show good improvements. And the improvements from the proposed claims are relatively smaller compared to the gains from directly using synthetic samples from SD without these proposed changes, as in Table 3. It then raises the question on the effectiveness of the propose method and the claimed contributions.

Introduction of additional information from SD. In Table 1 and 2, it shows that the classification performance does increase by adding synthetic samples following the proposed method. However, it is unclear that if the gain is indeed coming from these synthetic augmentation or just distilled information from the diverse datasets which the image generator Stable Diffusion is trained over.

**Questions:**

Can you explain the difference of the proposed marginal sample selection algorithm to Focal loss which also adjusts the sample weights based on the difficulty of the sample indicated by the raw softmax score of the classifier being learnt.

Line 13 of algorithm1, S_f is not defined in section 3.4, pls explain.

---

> ### Author Response · Authors · 2024-11-26
> **Response to Reviewer Ln52 (part 1)**
>
> `Q-1:` Lack of technical novelty. Though the motivation of the idea makes sense, my biggest concern is on the technical novelty of the proposed algorithm. The proposed design components on ensuring sample diversity, generation distribution alignment with real data distribution, and the marginal sample selection are all coming from existing literature or prior works.
>
> `A-1:` The challenge of effectively leveraging large-scale synthetic data to enhance downstream tasks remains a significant open problem, and **a general solution or pipeline for addressing this issue is still underexplored**. The proposed GenDataAgent offers such a pipeline, addressing key challenges like ensuring diversity, maintaining alignment with real data distributions, and targeting marginal samples as generator feedback.
>
> Although the individual components (e.g., LLM caption perturbation, VoG filtering, and marginal sample targeting) are grounded in existing literature, **their application in GenDataAgent is novel in both purpose and integration**. For example:
> - LLMs are commonly used for generating human-like conversational responses, but we repurpose LLMs to perturb image captions, thereby enhancing the diversity of prompts and, in turn, the synthetic data distribution.
> - In prior works, marginal samples—those hard to classify—are typically handled with weighted loss adjustments. In contrast, we utilize these samples as direct feedback to improve the generation process, enabling better alignment between synthetic and real data distributions.
>
> `Q-2:` Lack of sufficient experiment results to validate the effectiveness of the proposed methods and the claims in the contributions. The contribution claims the effectiveness of the proposed method from the three aspects of marginal sample selection, sample diversity and the distribution alignment to real data distribution. However, from the ablation study, it appears that even without these proposed improvements, the classification model trained with synthetic data also show good improvements. And the improvements from the proposed claims are relatively smaller compared to the gains from directly using synthetic samples from SD without these proposed changes, as in Table 3. It then raises the question on the effectiveness of the propose method and the claimed contributions.
>
> `A-2:`
> 1. GenDataAgent builds on the SOTA Real-Fake method, which already achieves saturated accuracy in the “synthetic data augmentation” setting. This makes further improvements challenging. However, our techniques demonstrate **larger performance gains in the “synthetic data only” setting**, as evidenced in Table 1, where the baseline's reliance on real data is reduced, and the proposed methods prove their effectiveness.
>
> 2. To further address the concerns about scalability and robustness, we conducted a scaling ablation study, varying the ratio of synthetic to real data (#synthetic data = 10x #real data and other ratios), as shown in Figure 4. Notably, Real-Fake's performance deteriorates when the volume of synthetic data increases across most datasets. In contrast, GenDataAgent remains stable, indicating its ability to handle varying synthetic data ratios effectively. This highlights a critical limitation of **existing methods** like Real-Fake: their **inability to maintain performance when scaling synthetic data**. **Our proposed techniques** mitigate this limitation, **ensuring consistent performance improvements even as synthetic data scales**.
>
> `Q-3:` Introduction of additional information from SD. In Table 1 and 2, it shows that the classification performance does increase by adding synthetic samples following the proposed method. However, it is unclear that if the gain is indeed coming from these synthetic augmentation or just distilled information from the diverse datasets which the image generator Stable Diffusion is trained over.
>
> `A-3:`
> 1. We think the gain comes from both aspects, and they cannot be disentangled when using synthetic data augmentation. For one thing, the synthetic data augments the real data and thus enlarges the training data volume. For another, the synthetic data are not supposed to provide totally the same information as real ones. To make the augmentation meaningful, the synthetic data needs to contain some attributes that the real one doesn’t have.
> 2. In the SD scenario, we adapt SD to target data distribution by LoRA fine-tuning, yet the generated images still benefit from the generating ability of SD, i.e., the diverse datasets which SD is trained over. Even though the performance gain mostly comes from distilling information from SD, this approach is also effective as it avoids injecting massive unrelated data into the downstream task.

---

> ### Author Response · Authors · 2024-11-26
> **Response to Reviewer Ln52 (part 2)**
>
> `Q-4:` Can you explain the difference of the proposed marginal sample selection algorithm to Focal loss which also adjusts the sample weights based on the difficulty of the sample indicated by the raw softmax score of the classifier being learnt.
>
> `A-4:`
> 1. **Focal loss** assigns a weighted loss based on the difficulty of the sample by introducing a modulating factor $(1-p_t)^{\gamma}$, which aims to down-weight the loss of "easy" examples. It **focuses on a very im-balance dataset and directly impacts the training process**.
> 2. First, **GenDataAgent does not require any assumption of the target dataset**. Second, the marginal sample selection picks challenging examples and uses them to guide the generator. After generating synthetic data, real and synthetic data are treated equally. The data distribution is transparent, so we **do not assign different weights to the data**. There, the difficulty level used in marginal sample selection **does not directly affect the training**.
>
> `Q-5:` Line 13 of algorithm1, S_f is not defined in section 3.4, pls explain.
>
> `A-5:` Thank you for pointing it out. $S_{f,j}$ is defined in L267, which is a subset of synthetic dataset $S_c$ with minimum sum of VoG scores. We refine this formula by adding the constraint to the number of filtering samples, highlighted in red color.

---

> ### Author Response · Authors · 2024-12-01
> **Further discussion with Reviewer Ln52**
>
> Dear Reviewer Ln52,
>
> We sincerely appreciate your time and effort in reviewing our manuscript. We have carefully reviewed and responded to the questions and concerns raised in your initial review:
>
> 1. We clarify that the novelty of our method lies in proposing a general pipeline for focused, diverse, and in-distribution synthetic data augmentation, which is underexplored in previous works [1,2,3,4].
> 2. Regarding the experiments, we demonstrated a significant performance gain in the "synthetic data only" setting (Table 1), as well as in the scaling ablation of synthetic data volume (Figure 4).
> 3. Additionally, we provide clarification on the role of SD, the difference between our marginal sample selection and Focal loss, and have refined the notation in the manuscript to improve clarity.
>
> As the author-reviewer discussion phase is drawing to a close, we would like to confirm whether our responses have adequately addressed your concerns. We are pleased to report that all other reviewers (NMJi, rGcL, and rpaL) indicated that their concerns have been satisfactorily resolved. If you still have any remaining questions or if any parts of our work need further clarification, please do not hesitate to let us know.
>
> Thank you in advance for your help.
>
> Sincerely,
>
> Authors
>
> [1] Fake it till you make it: Learning transferable representations from synthetic imagenet clones. CVPR, 2023.
>
> [2] Image captions are natural prompts for text-to-image models. arxiv, 2023.
>
> [3] Is synthetic data from generative models ready for image recognition? ICLR, 2023.
>
> [4] Real-fake: Effective training data synthesis through distribution matching. ICLR, 2024.

---

> > ### Author Response · Authors · 2024-12-03
> > **Second call for discussion with Reviewer Ln52**
> >
> > Dear Reviewer Ln52,
> >
> > We sincerely appreciate your time and effort in reviewing our submission.
> >
> > In our rebuttal, we have provided extensive analyses, explanations, and clarifications. We believe we have addressed your concerns well. We tried to get a response from you by posting some messages but got no answer. The Discussion Period is approaching its close. We call for a discussion with you once again. Could you please read our rebuttal and give a response? Thanks.
> >
> > Sincerely,
> >
> > Authors

---

### Author Response · Authors · 2024-11-26
**Response to all reviewers and area chairs for a brief summary**

Dear reviewers and area chairs,

We sincerely thank all reviewers and area chairs for their valuable time and insightful comments.

We are pleased to note that:

1. Reviewer Ln52 acknowledges that the **motivation behind our method makes sense**, both reviewers rGcL and NMJi agree that our approach demonstrates **a clear motivation**, and reviewer rpaL describes our approach as **a significant advancement** in training dataset augmentation.
2. Reviewer rGcL commends the **detailed analysis** of the proposed framework, and reviewer NMJi highlights that our GenDataAgent effectively integrates **a series of reasonable treatments**.
3. All reviewers recognize the **accuracy improvements** achieved by our method, with reviewer rpaL further emphasizing its **enhancement of robustness**.

We have responded to each reviewer individually and would like to summarize our responses here:

1. We clarify the **novelty** of the proposed framework in terms of both purpose and integration.
2. We demonstrated the **improvement** brought by our method through a scaling ablation study.
3. We addressed concerns about the training overhead and introduced two **lightweight versions for a better accuracy-time trade-off**.
4. We highlighted that our method can operate in **offline mode**, improving efficiency and enabling the reuse of synthetic data.
5. We added a detailed discussion comparing marginal sample selection and active learning.
6. We expanded on **related work, particularly [a]**, which incorporates a domain adaptation (DA) technique.
7. We conducted additional experiments incorporating **robustness metrics**, showing that our method outperforms the state-of-the-art Real-Fake baseline.
8. We clarified that our approach is **generalizable to other TTI methods and CV tasks**.
9. We presented additional results demonstrating that our method **avoids overfitting** compared to Real-Fake.
10. We provided further details in the paper, including the function of SD, the distinction between marginal sample selection and Focal loss, the use of consistent prompts for SD, and the choice of VoG checkpoints.

Again, we extend our gratitude to all reviewers and area chairs!

Best regards,

Authors

[a] A New Benchmark: On the Utility of Synthetic Data with Blender for Bare Supervised Learning and Downstream Domain Adaptation. CVPR, 2023.

---

### Meta-Review · Area_Chair_MvKh · 2024-12-20

**Metareview:**

This paper proposes the first of its kind method for on-the-fly synthetic data generation for data augmentation tailored to a specific downstream task using stable diffusion models. Specifically, it proposes a DataGenAgent pipeline to create synthetic data that is aligned to the distribution of the downstream AI task, while maintaining diversity and relevance of the data. The proposed method improves accuracy both in the synthetic data only and real + synthetic data training settings versus the state of the art.

**Additional Comments On Reviewer Discussion:**

Four reviewers, scored the paper as 3, 8, 8, 6. The reviewers initially raised concerns about the method using known prior techniques and hence not being adequately novel; and about its speed and generalization to other downstream tasks and OOD samples, among others. The majority of the reviewers' concerns were adequately addressed by the extensive and detailed responses provided by the authors during the rebuttal phase. The authors provided many convincing detailed discussion points and new experimental results for experiments requested by the reviewers. These resulted in three of the reviewers raising their scores towards recommending acceptance of the paper. Reviewer Ln52 was unresponsive. However, their concerns were shared by reviewer NMJi, which were successfully addressed by the authors' rebuttal responses.

All things considered, the AC agrees with the majority vote of the reviewers and recommends acceptance. It is recommended that the authors incorporate the changes that they have promised in the final version of their manuscript.

---

### Decision · Program_Chairs · 2025-01-22

Accept (Poster)